# Ozone enhancement due to photo-disassociation of nitrous acid in eastern China

**Xuexi Tie[1,2], Xin Long[1,5], Guohui Li[1], Shuyu Zhao[1], Junji Cao[1], Jianming Xu[3,4]**

[1] KLACP, SKLLQG, Institute of Earth Environment, Chinese Academy of Sciences, Xi'an 710061, China

[2] Center for Excellence in Urban Atmospheric Environment, Institute of Urban Environment, Chinese Academy of Sciences, Xiamen 361021, China

[3] Shanghai Meteorological Service, Shanghai, 200030, China

[4] Shanghai Key Laboratory of Meteorology and Health, Shanghai, 200030, China

[5] School of Environment Science and Engineering, Southern University of Science and Technology, Shenzhen 518055, China

*Correspondence to:* XueXi Tie (tiexx@ieecas.cn) or
Jianming Xu (metxujm@163.cn)

## Abstract

PM$_{2.5}$, a particulate matter with a diameter of 2.5 micrometers or less, is one of the major components of the air pollution in eastern China. In the past few years, China's government made strong efforts to reduce the PM$_{2.5}$ pollutions. However, another important pollutant (ozone) becomes an important problem in eastern China. Ozone (O$_3$) is produced by photochemistry, which requires solar radiation for the formation of O$_3$. Under heavy PM$_{2.5}$ pollution, the solar radiation is often depressed, and the photochemical production of O$_3$ is prohibited. This study shows that during late spring and early fall in eastern China, under heavy PM$_{2.5}$ pollutions, there were often strong O$_3$ photochemical productions, causing a co-occurrence of high PM$_{2.5}$ and O$_3$ concentrations. This co-occurrence of high PM$_{2.5}$ and O$_3$ is un-usual and is the main focus of this study. Recent measurements show that there were often high HONO surface concentrations in major Chinese mega cities, especially during daytime, with maximum concentrations ranging from 0.5 to 2 ppbv. It is also interesting to note that the high HONO concentrations were occurred during high aerosol concentration periods, suggesting that there were additional HONO surface sources in eastern China. Under the high daytime HONO concentrations, HONO can be photo-dissociated to be OH radicals, which enhance the photochemical production of O$_3$. In order to study the above scientific issues, a radiative transfer model (TUV; Tropospheric Ultraviolet-Visible) is used in this study, and a chemical steady state model is established to calculate OH radical concentrations. The calculations show that by including the OH production of the photo-dissociated of HONO, the calculated OH concentrations are significantly higher than the values without including this production. For example, by including HONO production, the maximum of OH concentration under the high aerosol condition (AOD=2.5) is similar to the value under low aerosol condition (AOD=0.25) in the no-HONO case. This result suggests that even under the high aerosol condition, the chemical oxidizing process for O$_3$ production can occurred, which explain the co-occurrence of high PM$_{2.5}$ and high O$_3$ in late spring and early fall seasons in eastern China. However, the O$_3$ concentrations were not significantly affected by the appearance of HONO in winter. This study shows that the seasonal variation of solar radiation plays important roles for controlling the OH production in winter. Because the solar radiation is in a very low level in winter, adding the photolysis of HONO has smaller effect in winter than in

other seasons, and OH remains low values by including the HONO production term.
This study provides some important scientific highlights to better understand the $O_3$
pollutions in eastern China.

**Keywords; High $PM_{2.5}$ and $O_3$, eastern China, HONO photolysis**


## 1. Introduction

Currently, China is undergoing a rapid economic development, resulting in a higher demand for energy and greater use of fossil fuels. As a result, the high emissions of pollutants produce heavy pollutions in mega cities of eastern China, such as Beijing and Shanghai. For example, in the city of Shanghai (a largest mega city in China), the urban and economical developments of the city are very rapid. During 1990 to 2015, the population increased from 13.3 to 24.1 million. The number of automobiles increased from 0.2 million (1993) to 2.0 million (2011). The rapid growing population and energy usage caused a rapid increase in the emissions of pollutants, leading to severe air pollution problems in these mega cities (Zhang et al., 2006; Geng et al., 2007; Deng et al., 2008).

Measurements, such as satellite observations have revealed much higher aerosol pollution in eastern China than in eastern US (Tie et al., 2006). The high aerosol pollution causes a wide range of environmental consequences. Jia et al. (2019) studied Anthropogenic Aerosol Pollution over the Eastern Slope of the Tibetan Plateau, and Zhu et al (2018) studied the impact of smoke aerosols from Russian forest fires on the air pollution over Asia. According to a study by Tie et al. (2009a), exposure to extremely high particle concentrations leads to a great increase of lung cancer cases. High PM (particular matter) concentrations also significantly reduce the range of visibility in China's mega cities (Deng et al., 2008). According to a recent study, the high aerosol pollution causes important effects on the crop (rice and wheat) production in eastern China (Tie et al., 2016).

In the troposphere, ozone formation is resulted from a complicated chemical process, and requires ozone precursors, such as VOCs (volatile organic carbons) and $NO_X = NO + NO_2$ (nitrogen oxides) (Sillman, 1995). As the increase in industrial activity and number of automobiles, the precursors of ozone ($O_3$) and the global budget of oxidization are also significantly increased (Huang et al., 2017; Huang et al., 2018). As a result, $O_3$ pollution becomes a serous pollution problem in Shanghai and other Chinese mega cities (Geng et al., 2010; Tie 2009b; Tie et al., 2015). The effects on $O_3$ production rate can be characterized as either $NO_X$-sensitive or VOC-sensitive

conditions. For the city areas, $O_3$ production is generally VOC-sensitive, while in the
remote area, $O_3$ production is generally NOx-sensitive in eastern China (Sillman,
1995; Zhang et al., 2003; Lei et al., 2004; Tie et al., 2013). Thus, better understanding
the trends of $O_3$ precursors (VOCs, $NO_X$) is important to determine the $O_3$ trends in
Shanghai (as well as many large cities in China).
In the past few years, China's government made strong efforts to reduce the $PM_{2.5}$
pollutions. However, another important pollutant ($O_3$) becomes an important problem
in eastern China. Several studies regarding the $O_3$ formation are previously studied in
Shanghai. For example, Geng et al. (2007; 2008) study the relationship between $O_3$
precursors (NOx and VOCs) for the ozone formation in Shanghai. Tie et al. (2009)
study the short-term variability of $O_3$ in Shanghai. Their study suggested that in
addition to the ozone precursors, meteorological conditions, such as regional transport,
have also strong impacts on the ozone concentrations. During September 2009, a
major field experiment (the MIRAGE-Shanghai) was conducted in Shanghai, and
multiply chemical species were measured during the experiment. The summary of the
measurements by Tie et al (2013) suggests that the ozone formation in Shanghai is
under VOC-sensitive condition. However, if the emission ration of NOx/VOCs
reduces to a lower value (0.1-0.2), the ozone formation in Shanghai will switch from
VOC-sensitive condition to NOx-sensitive condition.
Despite of some progresses have been made for the ozone formation in mega cities in
China, it is still lack of study of ozone development in large cities of China. For
example, this study shows that during late spring and early fall in eastern China,
under heavy $PM_{2.5}$ pollutions, there were often strong $O_3$ chemical productions,
causing the co-occurrence of high $PM_{2.5}$ and $O_3$ concentrations. Under heavy aerosol
condition, the solar radiation is depressed, significantly reducing the photochemical
production of $O_3$. This co-occurrence of high $PM_{2.5}$ and $O_3$ is an unusual and is the
focus of this study. He and Carmichael (1999) suggest that aerosol particles can
enhance the scattering of solar radiation, enhancing the flux density inside the
boundary layer. Recent measurements also show that there were often high HONO
concentrations in major Chinese mega cities, especially during daytime, with
maximum concentrations ranging from 0.5 to 2 ppbv (Huang et al., 2017). Zhang et al.
(2016) suggest that there are several potential HONO sources, including surface
emissions, conversion of $NO_2$ at the ocean surface, etc., and adding these sources can
improve the calculated HONO concentrations. It is also interesting to note that the
high HONO surface concentrations were occurred during high aerosol concentration
periods, suggesting that there are additional HONO surface sources in eastern China.
Under the high daytime HONO concentrations, HONO can be photo-dissociated to be
OH radicals, which enhance the photochemical production of $O_3$.

The paper is organized as follows: in Section 2, we describe the measurement of $O_3$
and $PM_{2.5}$. In Section 3, we describe the calculation of photo-dissociated rate of
HONO and a steady state model for the calculation of OH, and the causes of high $O_3$
production under the heavy aerosol condition. Section 4 shows a brief conclusion of
the results.
**2. Measurements of $O_3$ and $PM_{2.5}$**

There are long-term measurements in Eastern China by Chinese Environment
Protection Agency (CEPA) for monitoring the air quality in China. In eastern China,
especially in the capital city of China (Beijing), there are often heavy air pollutions,
especially for fine particular matter ($PM_{2.5}$ – the radium of particle being less than 2.5
um). Figure 1 shows the measurement sites in Beijing, in which the measured
concentrations of $PM_{2.5}$ and $O_3$ are used to the analysis. In the region, the air
pollutions were very heavy, especially in winter (Long et al., 2016; Tie et al., 2017).
The previous studies suggested that the both aerosol and $O_3$ pollutions became the
major pollutants in the region (Li et al., 2017).

Figure 2 shows the daily averaged concentrations of $PM_{2.5}$ and $O_3$ in the Beijing
region in 2015. The daily averaged concentrations show that there were strong daily
and seasonal variations for both the concentrations of $PM_{2.5}$ and $O_3$. Despite the daily
variation, the concentrations of $PM_{2.5}$ existed a strong seasonal variation. For example,
there were very high concentrations during winter, with maximum of ~300 $\mu g/m^3$.
While in summer, the maximum concentrations reduced to ~150 $\mu g/m^3$. The seasonal
variability of $O_3$ concentrations were opposite with the $PM_{2.5}$ concentrations, with
lower concentrations in winter (< 50 $\mu g /m^3$) and higher concentrations in summer (>
150 $\mu g/m^3$). These seasonal variations of $PM_{2.5}$ and $O_3$ have been studied by previous
studies (Tie and Cao, 2017; Li et al., 2017). Their results suggest that the winter high
$PM_{2.5}$ concentrations were resulted from the combination of both the high emissions
(heating season in the Beijing region), and poor meteorological ventilation conditions,
such as lower PBL (Planetary Boundary Layer) height (Quan et al., 2013; Tie et al.
2015). According to the photochemical theory of $O_3$ formation, the summer high and
winter low $O_3$ concentrations are mainly due to seasonal variation of the solar
radiation (Seinfeld, J. H. and Pandis, 2006).

The heavy aerosol concentrations play important roles to reduce solar radiation,
causing the reduction of $O_3$ formation. (Bian et al., 2007). As we show in Fig. 3
(upper panel), during wintertime, the $O_3$ concentrations were strong anti-correlated
with the $PM_{2.5}$ concentrations, suggesting that the reduction of solar radiation by
aerosol particles have important impact on the reduction of $O_3$ concentrations. Figure
3 (upper panel) also shows that the relationship between $O_3$ and $PM_{2.5}$ was not
linearly related. For example, when the concentrations of $PM_{2.5}$ were less than 100
$\mu g/m^3$, $O_3$ concentrations rapidly decreased with the increase of $PM_{2.5}$ concentrations.
In contrast, when the concentrations of $PM_{2.5}$ were greater than 100 $\mu g/m^3$, $O_3$
concentrations slowly decreased with the increase of $PM_{2.5}$ concentrations. This is
consistent with the result of Bian et al (2007).

It is interesting to note that from late spring to early fall periods, the correlation
between $PM_{2.5}$ and $O_3$ concentrations was positive relationship compared to the
negative relationship in winter (see Fig. 3 (lower panel)). This result suggests that $O_3$
production was high during the heavy haze period, despite the solar radiation was
greatly depressed. In order to clearly display this unusual event, we illustrate diurnal
variations of $PM_{2.5}$ and $O_3$, and $NO_2$ during a fall period (from Oct.5 to Oc. 6, 2015).
Figure 4 shows that during this period (as a case study), the $PM_{2.5}$ concentrations were
very high, ranging from 150 to 320 $\mu g/m^3$. Under such high aerosol condition, the
solar radiation should be significantly reduced, and $O_3$ photochemical production
would be reduced. However, the diurnal variation of $O_3$ was unexpectedly strong,
with high noontime concentration of $>220$ $\mu g/m^3$ and very low nighttime
concentration of $\sim25$ $\mu g/m^3$. This strong diurnal variation was due to the
photochemical activity, which suggested that during relatively low solar conditions,

the photochemical activities of $O_3$ production was high. According to the theory of the $O_3$ chemical production, the high $O_3$ production is related to high oxidant of OH (Seinfeld and Pandis, 2006), which should not be occurred during lower solar radiation. This result brings important issue for air pollution control strategy, because both $PM_{2.5}$ and $O_3$ are severe air pollutants in eastern China.

To clearly understand the effect of the high aerosol concentrations on solar radiation, we investigate the meteorological conditions, such as cloud covers, relation humidity (RH), and solar radiation during the period of the case study (see Figs. 5 and 6). Figure 5 shows that the cloud condition was close to the cloud free condition,but there was a very heavy aerosol layer in the Beijing region, suggesting that cloud cover played a minor role in the reduction of the solar radiation. The measured RH values (not shown) were generally higher than 60%, with a maximum of 95% during the period. As a result, the high aerosol concentrations companied by high RH produced important effects on solar radiation. As shown in Fig. 6, the daytime averaged solar radiation was significantly reduced (about 40% reduction in Oct. 5-6 period compared with the value of Oct. 8).

## 3. Method

In order to better understand the $O_3$ chemical production occurred in heavy aerosol condition in eastern China, the possible $O_3$ production in such condition is discussed. Ozone photochemical production ($P[O_3]$) is strongly related to the amount of OH radicals (Chameides et al., 1999). According to the traditional theory, the amount of surface OH radicals is proportional to the surface solar radiation, which is represented by

$$[OH] = P[HOx]/L[HOx]^* \qquad \text{(R-1)}$$

Where [OH] represents the concentration of hydroxyl radicals ($\#/cm^3$); HOx represents the concentration of $HO_2$ + OH ($\#/cm^3$); P[HOx] represents the photochemical production of HOx ($\#/cm^3/s$); and L[HOx]* (1/s) represents the photochemical destruction of HOx, which is normalized by the concentrations of OH.

The major process for the photochemical production of P[HOx] is through the $O_3$
photolysis and follows by the reaction with atmospheric water vapor. It can be
expressed as
$P[HOx] = J_1[O_3]/(k_1 \times am) \times 2.0 \times k_2[H_2O] = P_1[HOx]$          (R-2)

Where $J_1$ represents the photolysis of $O_3 + hv \rightarrow O^1D$; $k_1$ represents the reaction rate
of $O^1D + am \rightarrow O^3P$; and $k_2$ represents the reaction rate of $O^1D + H_2O \rightarrow 2OH$. As
we can see, this HOx production is proportional to the magnitude of solar radiation
$(J_1)$, and $J_1$ is the $O_3$ photolysis with the solar radiation. Figure 7 shows the
relationship between the values of $J_1$ and aerosol concentrations in October at
middle-latitude calculated by the TUV model (Madronich and Flocke, 1999). This
result suggests that under the high aerosol concentrations (AOD = 2.5), the $J_1$ value is
strongly depressed, resulting in significant reduction of OH concentrations and $O_3$
production. For example, the maximum $J_1$ value is about $2.7 \times 10^{-5}$ (1/s) with lower
aerosol values (AOD = 0.25). According to the previous study, the surface $PM_{2.5}$
concentrations were generally smaller than 50 $\mu g/m^3$ with this AOD value (Tie et al.,
2017). However, when the AOD value increase to 2.5 (the $PM_{2.5}$ concentrations are
generally >100 $\mu g/m^3$), the maximum $J_1$ value rapidly decreases to about $6 \times 10^{-6}$ (1/s),
which is about 450% reduction compared to the value with AOD=0.25. This study
suggests that under high $PM_{2.5}$ concentrations (>100 $\mu g/m^3$), the photochemical
production of OH (P[HOx]) is rapidly decreased, leading to low OH concentrations,
which cannot initiate the high oxidation of $O_3$ production. As a result, the high $O_3$
production shown in Fig. 4 cannot be explained. Other sources for $O_3$ oxidation are
needed to explain this result.

Recent studies show that the HONO concentrations are high in eastern China (Huang
et al., 2017). Because under high solar radiation, the photolysis rate of HONO is very
high, resulting in very low HONO concentrations in daytime (Seinfeld and Pandis,
2006). These measured high HONO concentrations are explained by their studies.
One of the explanations is that there are high surface HONO sources during daytime,
which produces high HONO concentrations (Huang et al., 2017). Zhang et al. (2016)
suggest that there are several potential HONO sources, including surface emissions,
conversion of $NO_2$ at the ocean surface, etc. Zhang et al. (2016) parameterized these

potential HONO sources in the WRF-Chem model, and the calculated HONO concentrations are increased in the WRF-Chem model.

The version of the WRF-Chem model is based on the version developed by Grell et al. (2015), and is improved mainly by Tie et al. (2017) and Li et al. (2011). The chemical mechanism chosen in this version of WRF-Chem is the RADM2 (Regional Acid Deposition Model, version 2) gas-phase chemical mechanism. For the calculation of HONO, only the gas-phase chemistry of OH+NO is included to calculate HONO concentrations. As shown in Fig. 8, the calculated HONO concentrations are significantly smaller than the measured HONO values in eastern China, suggesting that in addition to the gas-reaction, there are missing HONO sources (surface sources or others). Because these missing sources are not fully understood and large uncertainty is remained, in the following calculation, we compare the OH concentrations due to both calculated HONO (without the missing sources) and the measured HONO concentrations to illustrate the importance of these missing sources for the production of OH radicals and to suggest that further study to better understand the missing sources is an urgent scientific issue.

Figure 8 shows the measured HONO concentrations in three large cities in China (Shanghai, Xi'an, and Beijing) during fall and winter. It also shows the corresponding $PM_{2.5}$ and $O_3$ in the 3 cities (i.e., Fig. 8a for Beijing, Fig. 8b for Shanghai, and Fig. 8c for Xian). It shows that the measured HONO concentrations were high, ranging from sub-ppbv to a few ppbv, with higher values during morning, and lower values in daytime. The co-occurrences of high $PM_{2.5}$ and $O_3$ happened in the 3 cities. As a result, we think that the high HONO is a common event in large cities in eastern China, especially in daytime. This high HONO is also measured by previous studies (Zhang et al. 2016; Huang et al. 2017). In this study, we make an assumption that the co-occurrence between $O_3$ and $PM_{2.5}$ occurred under high HONO concentrations. We note that using this assumption may result in some uncertainties in estimating the effect of HONO on OH. For example, using the measured HONO in Xi'an and Beijing could produce 1-2 times higher OH production by photolysis of HONO than the result by using the data from Shanghai. In this case, we use the measured HONO from Shanghai to avoid the over estimate of the HONO effect, which can be considered as a low-limit estimation.

It is also interesting to note that the high HONO concentrations were occurred during high aerosol concentration periods. Figure 9 illustrates that when the $PM_{2.5}$ concentrations increased to 70-80 $\mu g/m^3$, and the HONO concentrations enhanced to 1.4-18 ppbv during September in Shanghai. This measured high HONO concentrations were significantly higher than the calculated concentrations (shown in Fig. 8), suggesting that some additional sources of HONO are needed. This result is consistent with the HONO measurements in other Chinese cities (Huang et al. 2017).

The high HONO concentrations in daytime become a significant source of OH radicals. As a result, the OH production rate (P[HOx]) can be written to the following reactions.

$$P_2[HOx] = J_2 \times [HONO] \tag{R-3}$$

$$P[HOx] = P_1[HOx] + P_2[HOx]$$

$$= J_1[O_3]/(k_1 \times am) \times 2.0 \times k_2[H_2O] + J_2 \times [HONO] \tag{R-4}$$

Because the chemical lifetime of OH is less than second, OH concentrations can be calculated according to equilibrium of chemical production and chemical loss. With the both OH chemical production processes, the OH concentrations can be calculated by the following equation (Seinfeld and Pandis, 2006).

$$P1 + P2 = L1 + L2$$

Where P1 and P2 are the major chemical productions, expressed in R-4, and L1 and L2 are the major chemical loss of OH, and represent by

$$L1: \quad OH + NO_2 \rightarrow HNO_3 \tag{R-5}$$

$$L2: \quad HO_2 + HO_2 \rightarrow H_2O_2 + O_2 \tag{R-6}$$

Under high NOx condition, such as in the large cities in eastern China, NOx concentrations were often higher to 50 ppbv (as shown in Fig. 4). As a result, the L1 term is larger than L2. The OH concentrations can be approximately expressed as

$[HO] = \{J_1[O_3]/(k_1 \times am) \times 2.0 \times k_2[H_2O] + J_2 \times [HONO]\}/$

$k_3[NO_2]$                                              (R-5)

Where $k_3$ is the reaction coefficient of $OH + NO_2 \rightarrow HNO_3$.

## 4. Result and analysis

### 4.1. OH productions in different HONO conditions

In order to quantify the individual effects of these two OH production terms (P1 and
P2) on the OH concentrations, the P1 and P2 are calculated under different daytime
HONO conditions (calculated low HONO and measured high HONO concentrations).
Figure 10 shows that under the low HONO condition, the P1 is significantly higher
than P2, and P2 has only minor contribution to the OH values. For example, the
maximum of P1 occurred at 13 pm, with a value of $65 \times 10^6$ #/cm$^3$/s. In contrast, the
maximum of P2 occurred at 10 am, with a value of $15 \times 10^6$ #/cm$^3$/s. However, under
high HONO condition, the P2 plays very important roles for the OH production. The
maximum of P2 occurred at 11 am, with a value of $350 \times 10^6$ #/cm$^3$/s, which is about
500% higher than the P1 value. It is important to note that this calculation is based on
the high aerosol condition (AOD = 2.5) in September. This result can explain the high
$O_3$ chemical production in Fig. 4.

### 4.2. OH in different aerosol conditions

In order to understand the effect of aerosol conditions, especially high aerosol
conditions, on the OH concentrations. Figure 11 shows the OH concentrations with
and without HONO production of OH. With including the HONO production (i.e.,
including P1 and P2), the calculated OH concentrations are significantly higher than
without including this production (i.e., only including P1). The both calculated OH
concentrations are rapidly changed with different levels of aerosol conditions. For
example, without HONO production, the maximum OH concentration is about
$7.5 \times 10^5$ #/cm$^3$ under low aerosol condition (AOD=0.25). In contrast, the maximum
OH concentration rapidly reduced to $1.5 \times 10^5$ #/cm$^3$ under high aerosol condition
(AOD=2.5), and further decreased to $1.0 \times 10^5$ #/cm$^3$ with the AOD value of 3.5. In
contrast, with including HONO production, the OH concentrations significantly

increased. Under higher aerosol condition (AOD=2.5), the maximum of OH concentration is about $7.5 \times 10^5$ #/cm$^3$, which is the same value under low aerosol condition in the no-HONO case. This result suggests that the measured high O$_3$ production occurred in the high aerosol condition is likely due to the high HONO concentrations in Shanghai.

### 4.3. Effects of clouds

Cloud cover can have very important impacts on the photolysis of HONO, which can affect the effect of HONO on the OH radicals. The above calculations are based on the cloud-free condition, with heavy aerosol concentration in the Beijing region. As shown in Fig. 5, during the case study period (Oct 5 to 6, 2015) (see Fig. 4), the weather map shows that the cloud-free condition, with heavy aerosol condition.

In order to understand the effects of cloud on the photolysis of HONO, we include different cloud covers in the TUV model. The calculated results show in Fig. 12. The results show that the thin cloud (with cloud cover in 2 km and cloud water of 10 g/m$^3$), could reduce the photolysis rate of HONO by about 40%, but the HONO could still remain important effects. However, with dense cloud condition (with cloud covers at 2 and 3 km and cloud water of 50 10 g/m$^3$), the photolysis rate of HONO could reduce by 9-10 times by the cloud. In this case, adding photolysis rate of HONO cannot produce important effect on OH radicals and the production of O$_3$.

### 4.4. OH in winter

The measurement of O$_3$ also shows that the concentrations in winter were always low (see Fig. 2), suggesting that the O$_3$ concentrations were not significantly affected by the appearance of HONO. Figure 13 shows the OH concentrations in September and December. It shows that under different aerosol conditions, OH concentrations in December were very low compared with the values in September. Both the calculated OH concentrations include the HONO production term. For example, under the condition of AOD=2.5, the maximum OH is about $7.5 \times 10^5$ #/cm$^3$ in September, while it rapidly reduces to $1.5 \times 10^5$ #/cm$^3$ in December. Under the condition of AOD=3.5, the maximum OH is still maintaining to a relative high level ($4.5 \times 10^5$ #/cm$^3$) in September. However, the maximum OH values are extremely low in December, with

maximum value of $0.5 \times 10^5$ #/cm$^3$ in December. Because both the OH chemical
productions (P1 and P2) are strongly dependent upon solar radiation (see equation
R-4), the seasonal variation of solar radiation plays important roles for controlling the
OH production in winter (see Fig. 13). Because the solar radiation is in a very low
level in winter, adding the photolysis of HONO has smaller effect in winter than in
other seasons and OH remains low values by including the HONO production term.

**Summary**

Currently, China is undergoing a rapid economic development, resulting in a high
demand for energy, greater use of fossil fuels. As a result, the high emissions of
pollutants produce heavy aerosol pollutions (PM$_{2.5}$) in eastern China, such as in the
mega city of Beijing. The long-term measurements show that in addition to the heavy
aerosol pollution, the O$_3$ pollution becomes another major pollutants in the Beijing
region. The measured results show that there were very strong seasonal variation in
the concentrations of both PM$_{2.5}$ and O$_3$ in the region. During winter, the seasonal
variability of O$_3$ concentrations were anti-correlated with the PM$_{2.5}$ concentrations.
However, from late spring to early fal, the correlation between PM$_{2.5}$ and O$_3$
concentrations was positive compared to the negative in winter. This result suggests
that during heavy aerosol condition (the solar radiation was depressed), the O$_3$
chemical production was still high, appearing a co-occurrence of high PM$_{2.5}$ and O$_3$ in
some cases from late spring to early fall. This co-occurrence of high PM$_{2.5}$ and O$_3$ is
the focus of this study. The results are highlighted as follows;

(1) There are high daytime HONO concentrations in major Chinese mega cities, such

as in Beijing and Shanghai. It is also interesting to note that the high HONO

concentrations were occurred during high aerosol concentration periods. Under

the high daytime HONO concentrations, HONO can be photo-dissociated to be

OH radicals, and becomes an important process to produce OH.

(2) With including the OH production of measured HONO concentrations, the

calculated OH concentrations are significantly higher than without including this

production. For example, without HONO production, the maximum OH

concentration is about $7.5 \times 10^5$ #/cm$^3$ under low aerosol condition (AOD=0.25),

and rapidly reduced to $1.5 \times 10^5$ #/cm$^3$ under high aerosol condition (AOD=2.5) in

September. In contrast, by including HONO production, the OH concentrations
significantly increased. For example, under higher aerosol condition (AOD=2.5),
the maximum of OH concentration is about $7.5\times10^5$ #/cm$^3$, which is similar to the
value under low aerosol condition in the no-HONO case. This result suggests that
even under the high aerosol conditions, the chemical oxidizing process for O$_3$
production can be active. This result is likely for explaining the co-occurrence of
high PM$_{2.5}$ and high O$_3$ from late spring to early in eastern China.
(3)  The measurement of O$_3$ also shows that the concentrations in winter were always
low, suggesting that the O$_3$ concentrations were not significantly affected by the
appearance of HONO. The calculated result shows that the seasonal variation of
solar radiation plays important roles for controlling the OH production in winter.
Because the solar radiation is in a very low level in winter, adding the photolysis
of HONO has smaller effect in winter than in other seasons, and OH remains low
values by including the HONO production term.
In recent years, the PM$_{2.5}$ pollutions are reduced due to the large control efforts by the
Chinese government, the O$_3$ pollutions become another severe pollution problem in
eastern China. This study is important, because it provides some important scientific
highlights to better understand the O$_3$ pollutions in eastern China.

**Data availability**. The data used in this paper can be provided upon request from
Xuexi Tie (tiexx@ieecas.cn).

**Author contributions**. XT came up with the original idea of investigating the
scientific issue. XT and JX designed the analysis method. XL, GL and SZ provided
the observational data and helped in discussion. XT prepared the manuscript with
contributions from all co-authors.

**Acknowledgement**
This work was supported by the National Natural Science Foundation of China
(NSFC) under Grant Nos. 41430424 and 41730108. The Authors thanks the supports
of Center for Excellence in Urban Atmospheric Environment, Institute of Urban
Environment, Chinese Academy of Sciences.

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

**Figure Caption**

**Fig. 1.** The geographic locations of the measurement sites in Beijing, in which the measured concentrations of $PM_{2.5}$ and $O_3$ are used to the analysis.

**Fig. 2.** The daily averaged concentrations of $PM_{2.5}$ and $O_3$ in the Beijing region in 2015. The concentrations are averaged over all sites shown in Fig. 1. The blue lines represent the $PM_{2.5}$ concentrations ($\mu g/m^3$), and the red bars represent the $O_3$ concentrations ($\mu g/m^3$). The rectangles show some typical events during winter (green), spring and fall (orange), and summer (red).

**Fig. 3.** The correlation between $O_3$ and $PM_{2.5}$ concentrations during winter (upper panel) and from late spring to eraly fall (lower panel). During winter, $O_3$ concentrations were strong anti-correlated with the $PM_{2.5}$ concentrations. From late spring to early fall, $O_3$ concentrations were correlated with the $PM_{2.5}$ concentrations.

**Fig. 4.** The diurnal variations of $PM_{2.5}$ (blue line) and $O_3$ (red line), and $NO_2$ (green line) during a fall period (from Oct.5 to Oc. 6, 2015). It shows that with high $PM_{2.5}$ condition, there was a strong $O_3$ diurnal variation.

**Fig. 5.** The cloud condition during the period of the case study (between Oct 5 and 6, 2015) in the Beijing region. The bright white color shows the cloud covers, and the grey white shows the haze covers. The Beijing region was under the heavy haze conditions during the period.

**Fig. 6.** The measured solar radiation ($W/m^2$) from Oct. 3 to Oct. 9, 2015 in Beijing. The upper panel shows hourly values, and the lower panel shows the daytime averaged values.

**Fig. 7.** The effect of aerosol levels with AOD = 0.25 (black line), AOD = 2.5 (red line), AOD = 3.5 (blue line), and AOD = 4.0 (green line) on the $O_3$ photolysis calculated by the TUV model in October at middle-latitude.

**Fig. 8a.** The measured HONO concentrations (ppbv) and the $PM_{2.5}$ and $O_3$ daily concentrations in Beijing. The upper panel shows the measured daily concentrations of $PM_{2.5}$ and $O_3$ as shown in Fig.2. The dark-red line was measured HONO in Beijing from 1 to 27 January, 2014.

**Fig. 8b.** The measured HONO concentrations (ppbv) and the $PM_{2.5}$ and $O_3$ daily concentrations in Shanghai. The upper panel shows the measured daily concentrations of $PM_{2.5}$ and $O_3$ in 2015. The dark-red line was measured in Shanghai from 9 to 18 September, 2009. The green line was calculated by the WRF-Chem model.

**Fig. 8c.** The measured HONO concentrations (ppbv) and the $PM_{2.5}$ and $O_3$ daily concentrations in Xi'an. The upper panel shows the measured daily concentrations of $PM_{2.5}$ and $O_3$ in 2015. The red line was measured HONO in Xi'An from 24 July to August 6, 2015.

**Fig. 9.** The measured HONO (upper left panel), $PM_{2.5}$ concentrations (lower left panel), and $O_3$ concentrations (upper right panel) in fall in Shanghai. It illustrates that

the high HONO concentrations were corresponded with high $PM_{2.5}$ concentrations.

**Fig. 10.** The calculated OH production P(HOx) (#/cm$^3$/s) by using the model
calculated HONO (low concentrations) (in the upper panel) and by using the
measured HONO (high concentrations) (in the lower panel). The red bars represent
the calculation of the P1 term, and the red bars represent the calculation of the P2
term   (OH production from HONO).

**Fig. 11.** The calculated OH concentrations (#/cm$^3$) with (upper panel) and without
(lower panel) HONO production of OH, under different aerosol levels. Dark red
(AOD=0.25), red (AOD=2.5) ), red (AOD=3.5) ), and red (AOD=4.0).

**Fig. 12.** The effect of cloud cover on the photolysis rate of HONO (J[HONO]). The
blue, red, and green lines represent the cloud water vapor of 0 (cloud-free), 10 (g/m$^3$ –
thin cloud), and 50 (g/m$^3$ – thick cloud), respectively. The left panel (A) represents
the light aerosol condition, with AOD of 0.25, and the right panel (B) represents the
heavy aerosol condition, with AOD of 2.5.

**Fig. 13.** The calculated OH concentrations in September (blue bars) and December
(dark red bars), under different aerosol levels.

## Figures

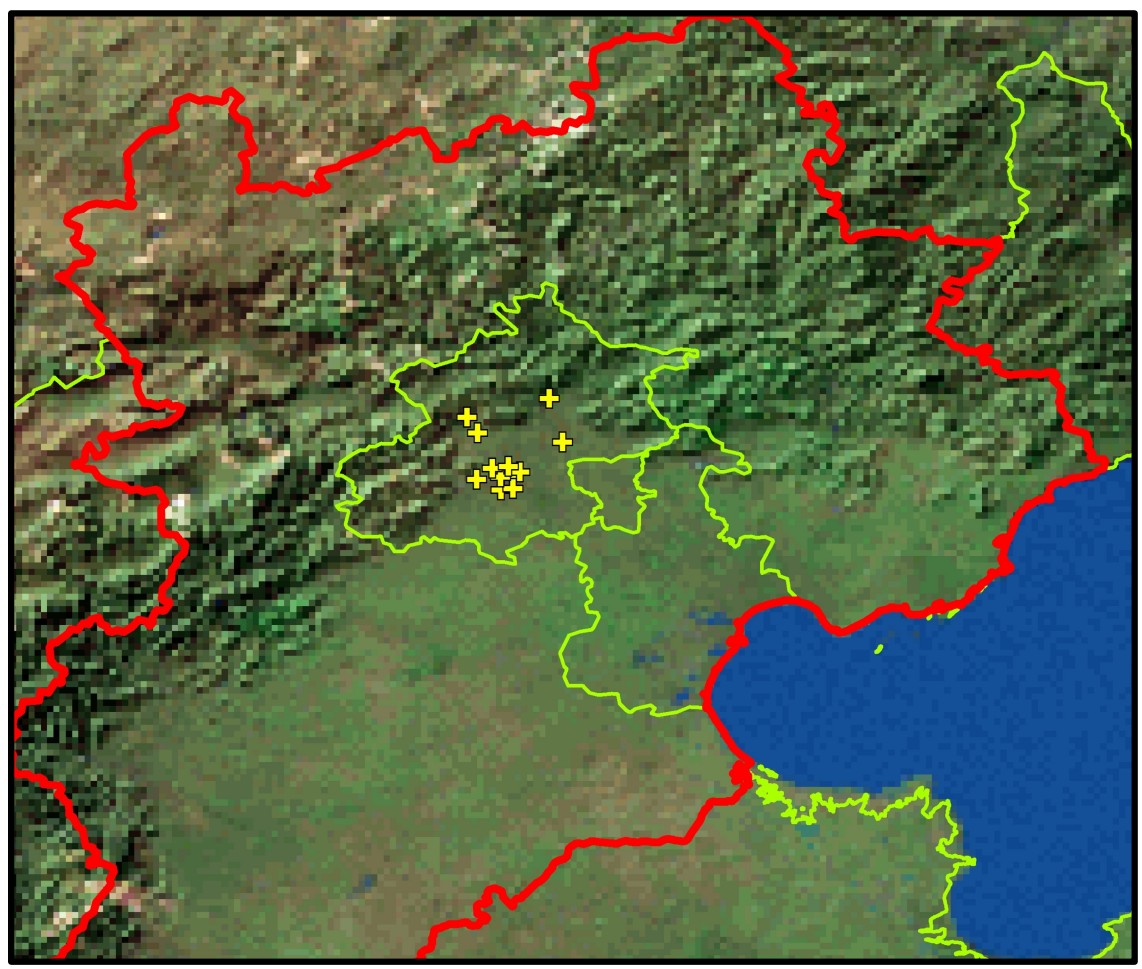

**Fig. 1.** The geographic locations of the measurement sites in Beijing, in which the measured concentrations of $PM_{2.5}$ and $O_3$ are used to the analysis.

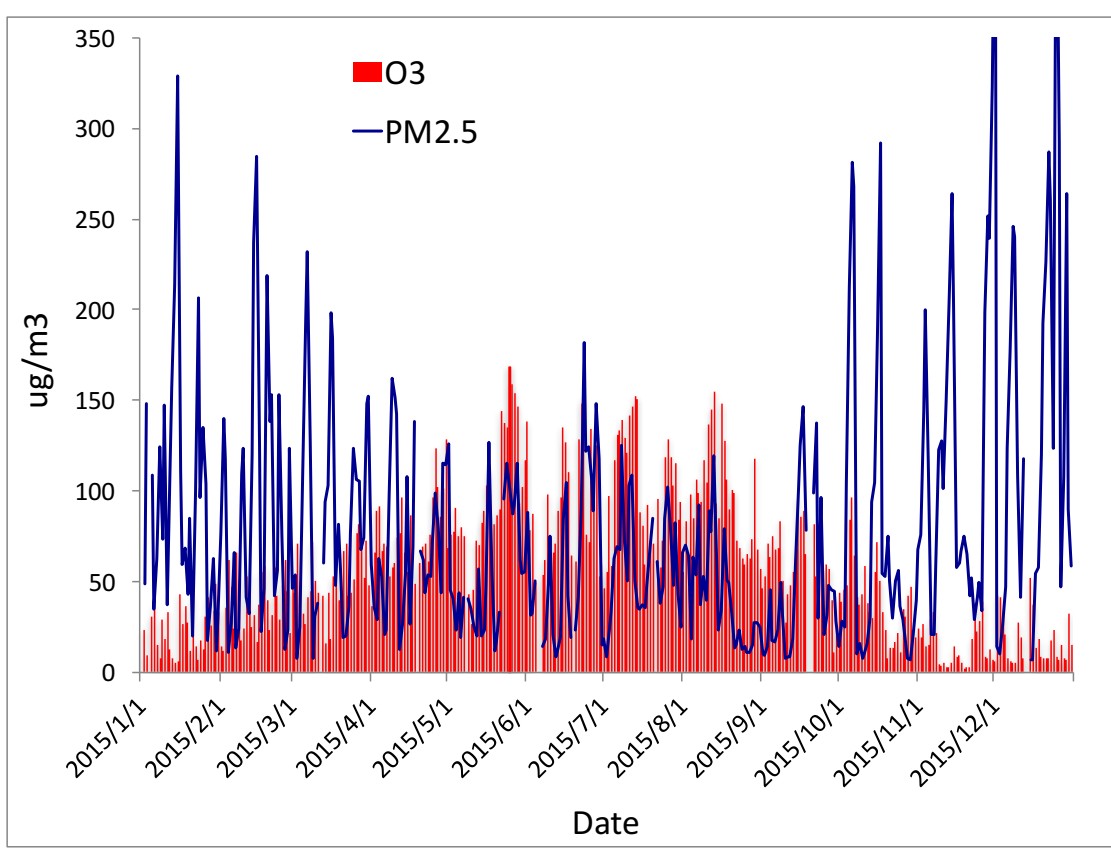

**Fig. 2.** The daily averaged concentrations of PM$_{2.5}$ and O$_3$ in the Beijing region in
2015. The concentrations are averaged over all sites shown in Fig. 1. The blue lines
represent the PM$_{2.5}$ concentrations ($\mu g/m^3$), and the red bars represent the O$_3$
concentrations ($\mu g/m^3$). The rectangles show some typical events during winter
(green), spring and fall (orange), and summer (red).


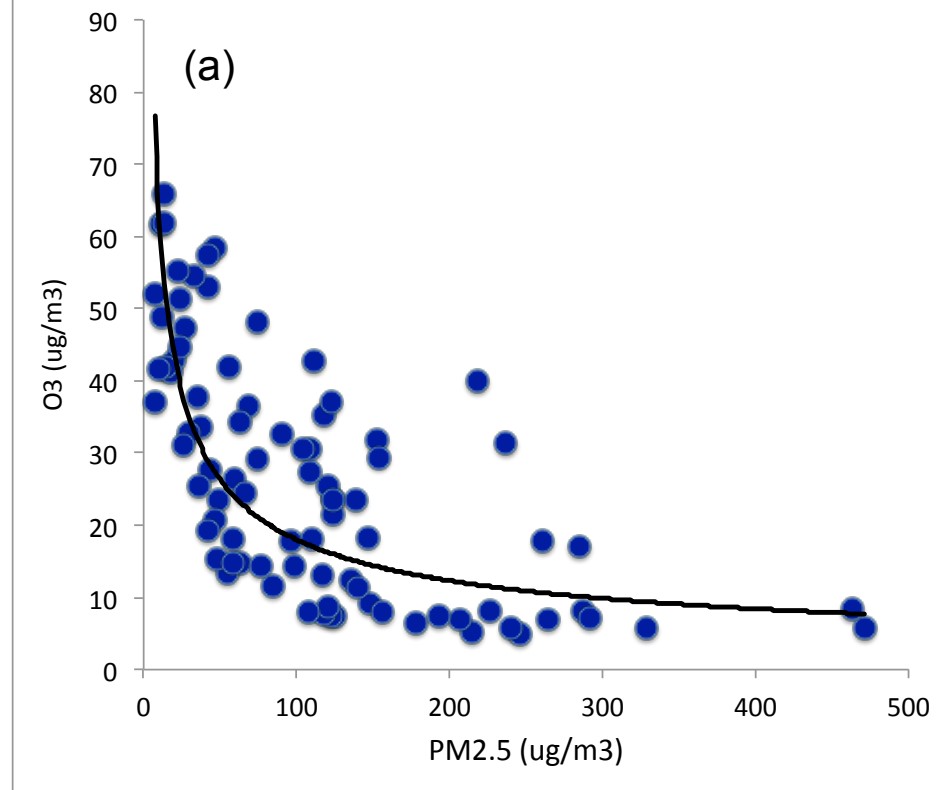


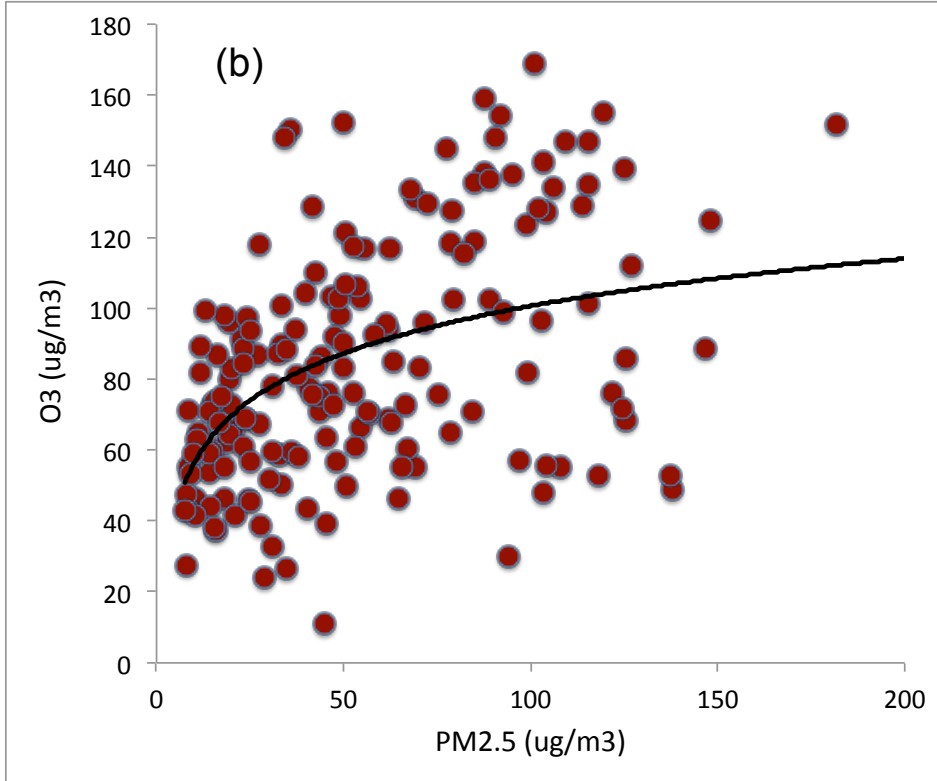

**Fig. 3.** The correlation between O$_3$ and PM$_{2.5}$ concentrations during winter (upper a)
and from late spring to early fall (panel b). During winter, O$_3$ concentrations were
strong anti-correlated with the PM$_{2.5}$ concentrations. From late spring to early fall, O$_3$
concentrations were correlated with the PM$_{2.5}$ concentrations.

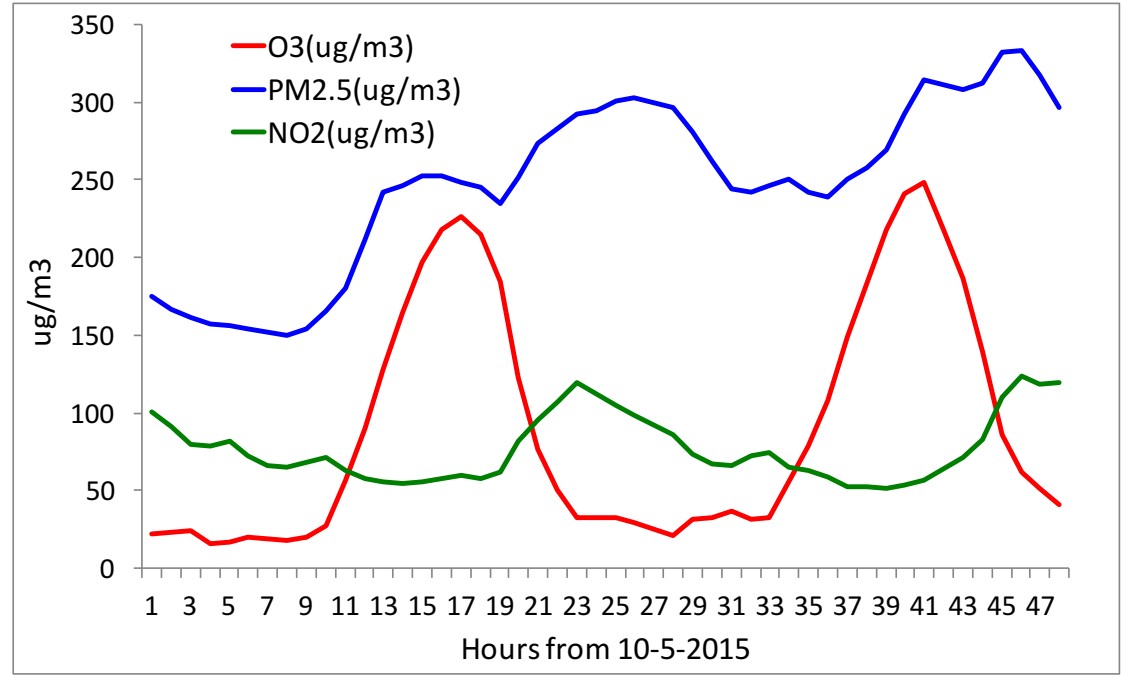

**Fig. 4.** The diurnal variations of PM$_{2.5}$ (blue line) and O$_3$ (red line), and NO$_2$ (green
line) during a fall period (from Oct. 5 to Oct. 6, 2015). It shows that with high PM$_{2.5}$
condition, there was a strong O$_3$ diurnal variation.

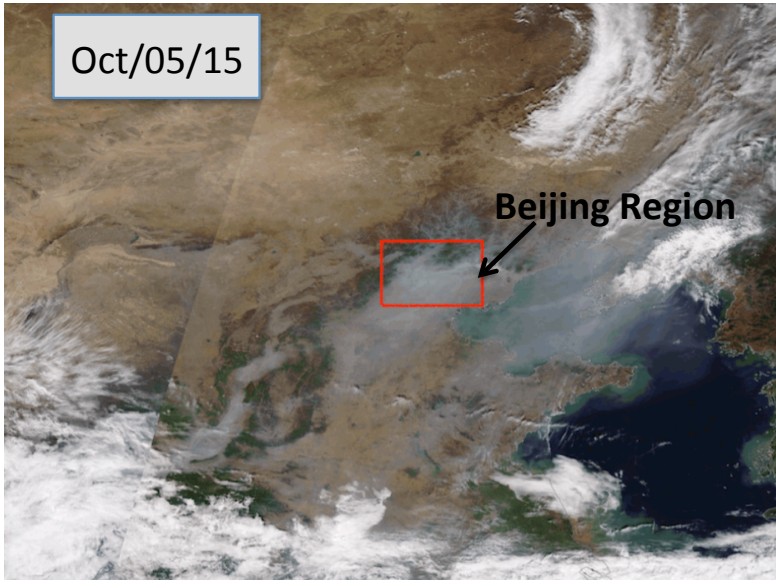

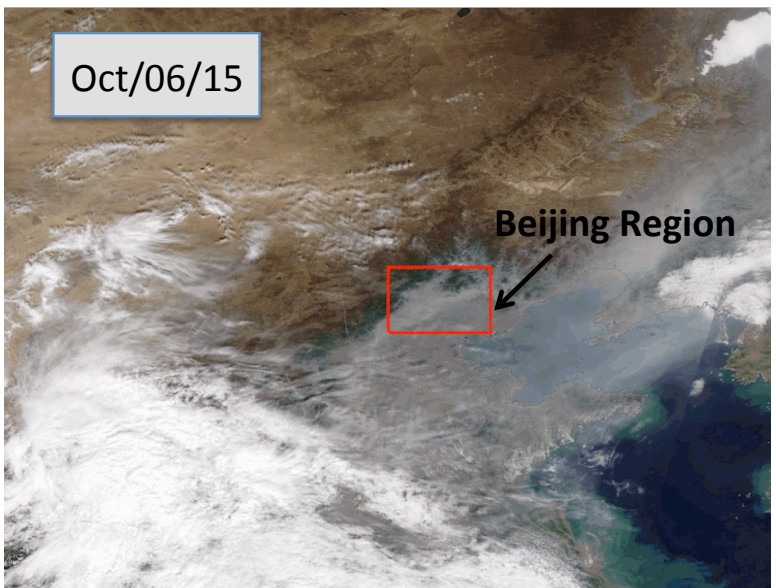

**Fig. 5.** The cloud condition during the period of the case study (between Oct 5 and 6,
2015 in the Beijing region. The bright white color shows the cloud covers, and the
grey white shows the haze covers. The Beijing region is under the heavy haze
conditions during the period.

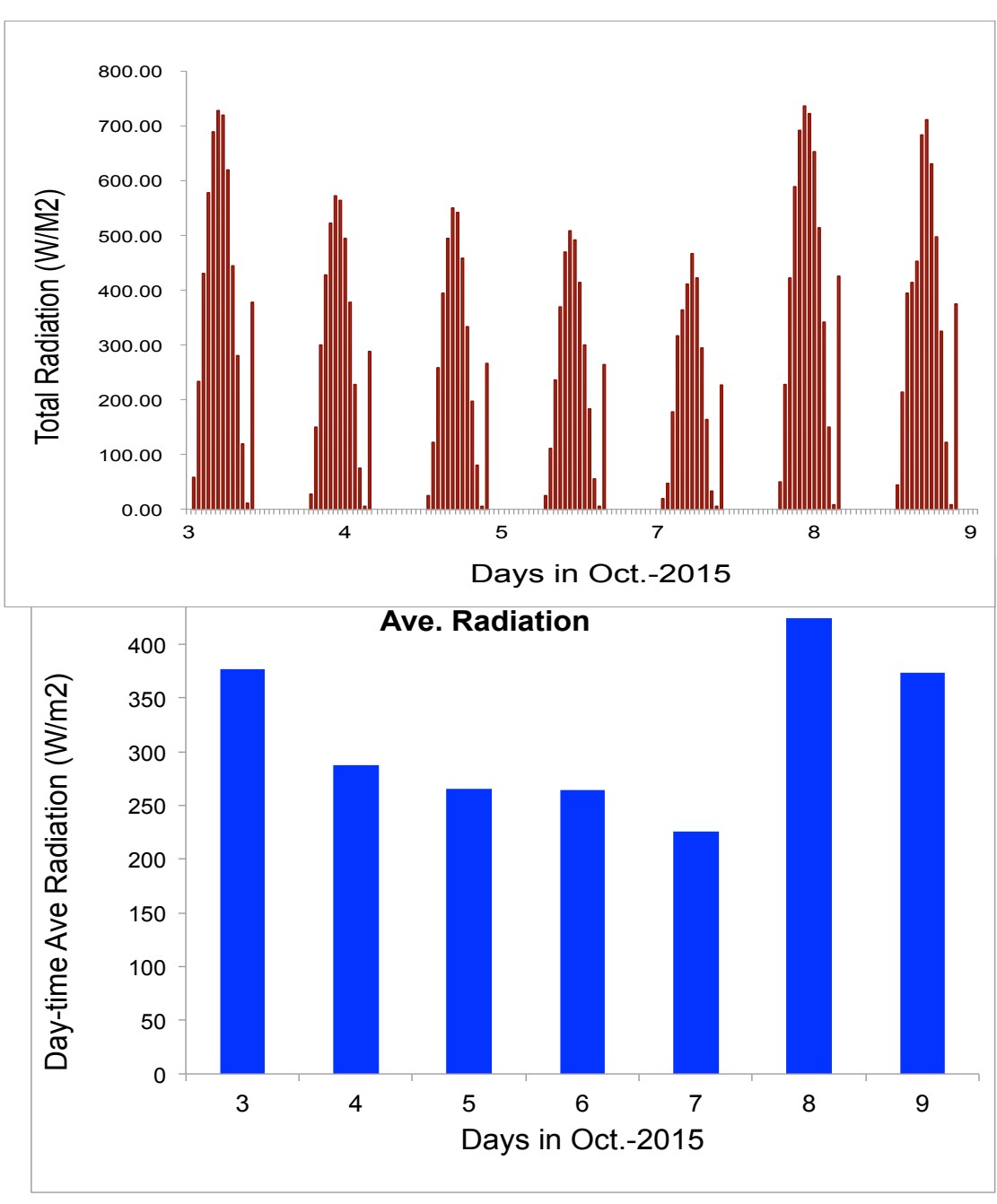

**Fig. 6.** The measured solar radiation (W/m²) from Oct. 3 to Oct. 9, 2015 in Beijing.
The upper panel shows hourly values, and the lower panel shows the daytime
averaged values.

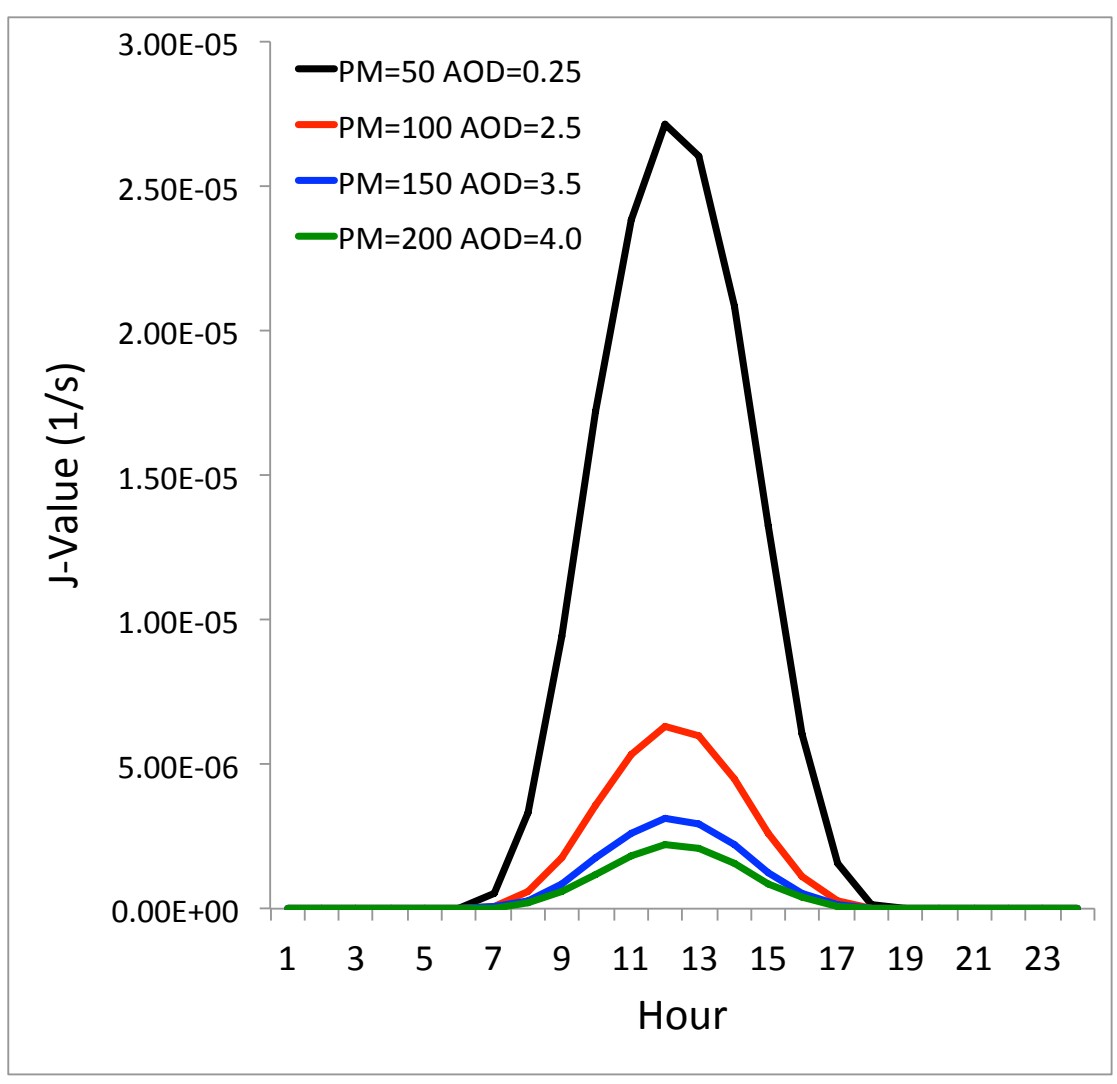

**Fig. 7.** The effect of aerosol levels with AOD = 0.25 (black line), AOD = 2.5 (red
line), AOD = 3.5 (blue line), and AOD = 4.0 (green line) on the $O_3$ photolysis
calculated by the TUV model in October at middle-latitude.

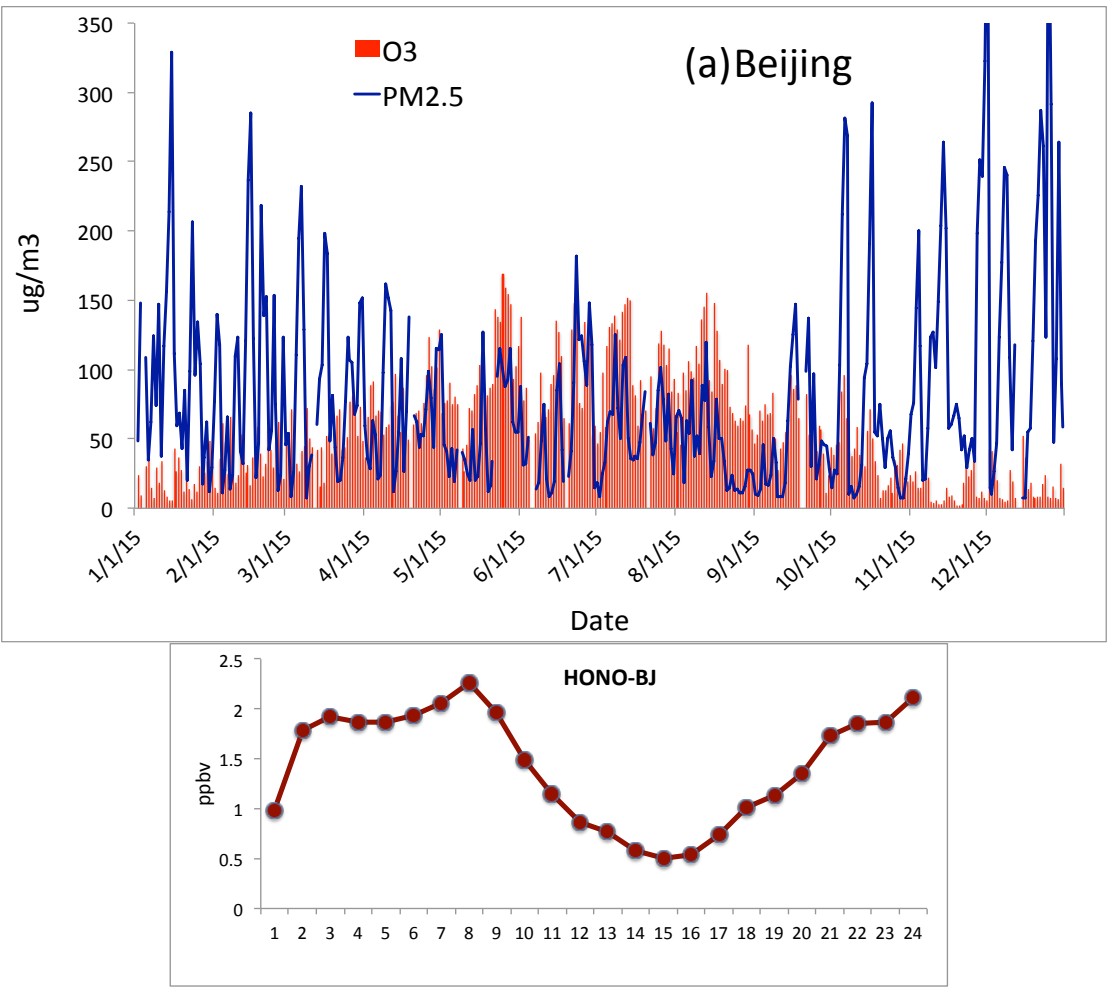

**Fig. 8a.** The measured HONO concentrations (ppbv) and the PM$_{2.5}$ and O$_3$ daily concentrations in Beijing. The upper panel shows the measured daily concentrations of PM$_{2.5}$ and O$_3$ as shown in Fig.2. The dark-red line was measured HONO in Beijing from 1 to 27 January, 2014.

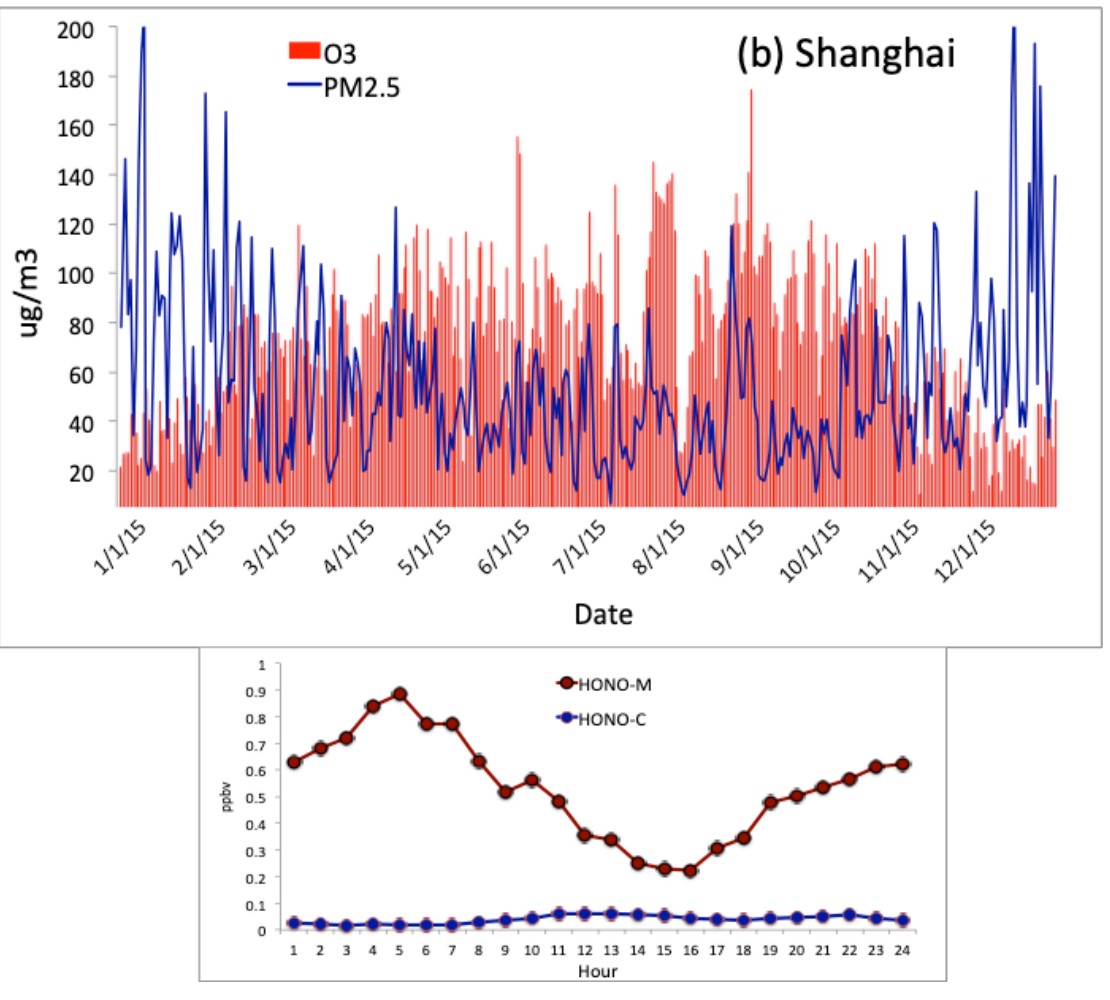

**Fig. 8b.** The measured HONO concentrations (ppbv) and the $PM_{2.5}$ and $O_3$ daily
concentrations in Shanghai. The upper panel shows the measured daily concentrations
of $PM_{2.5}$ and $O_3$ in 2015. The dark-red line was measured in Shanghai from 9 to 18
September, 2009. The green line was calculated by the WRF-Chem model.

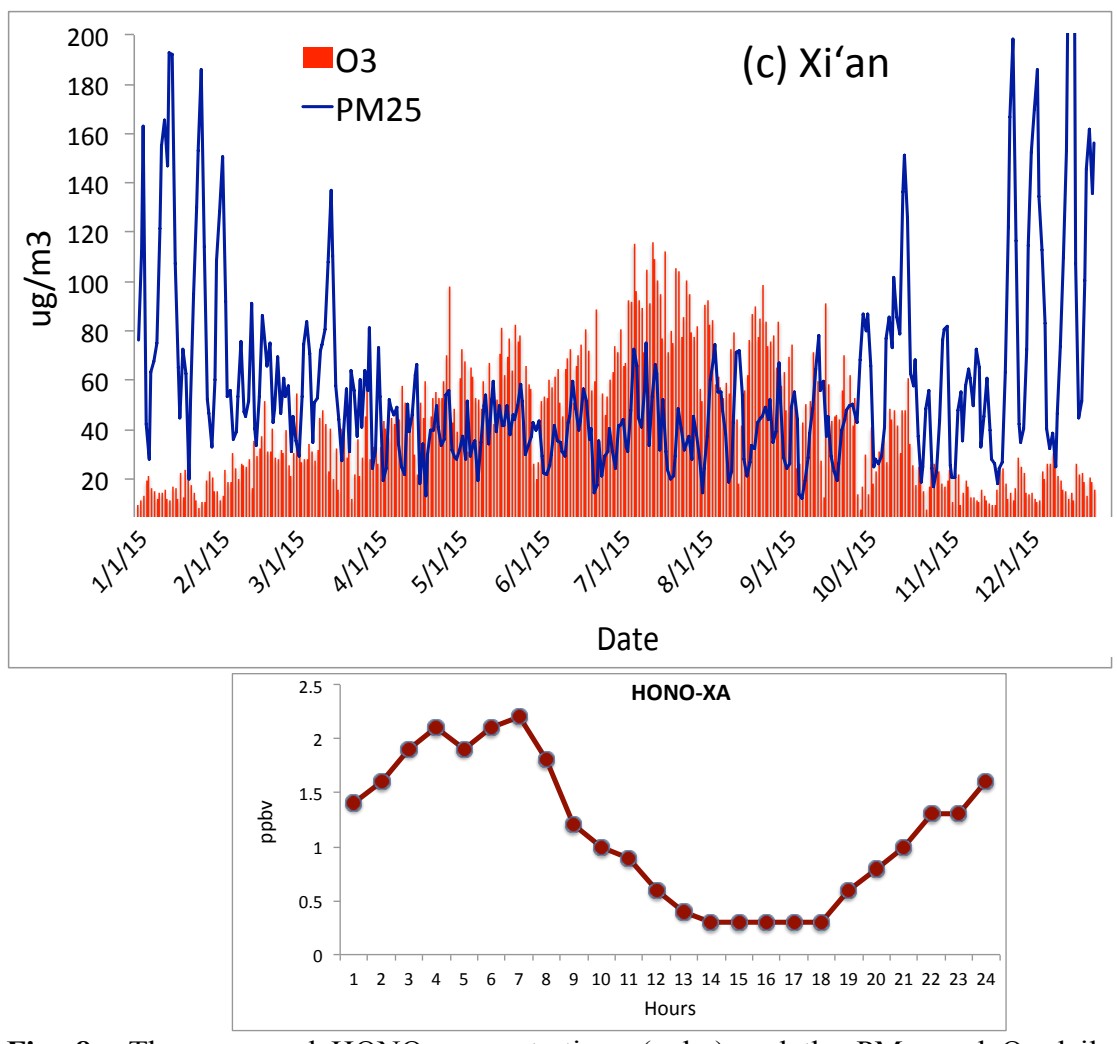

**Fig. 8c.** The measured HONO concentrations (ppbv) and the PM$_{2.5}$ and O$_3$ daily concentrations in Xi'an. The upper panel shows the measured daily concentrations of PM$_{2.5}$ and O$_3$ in 2015. The red line was measured HONO in Xi'An from 24 July to August 6, 2015.

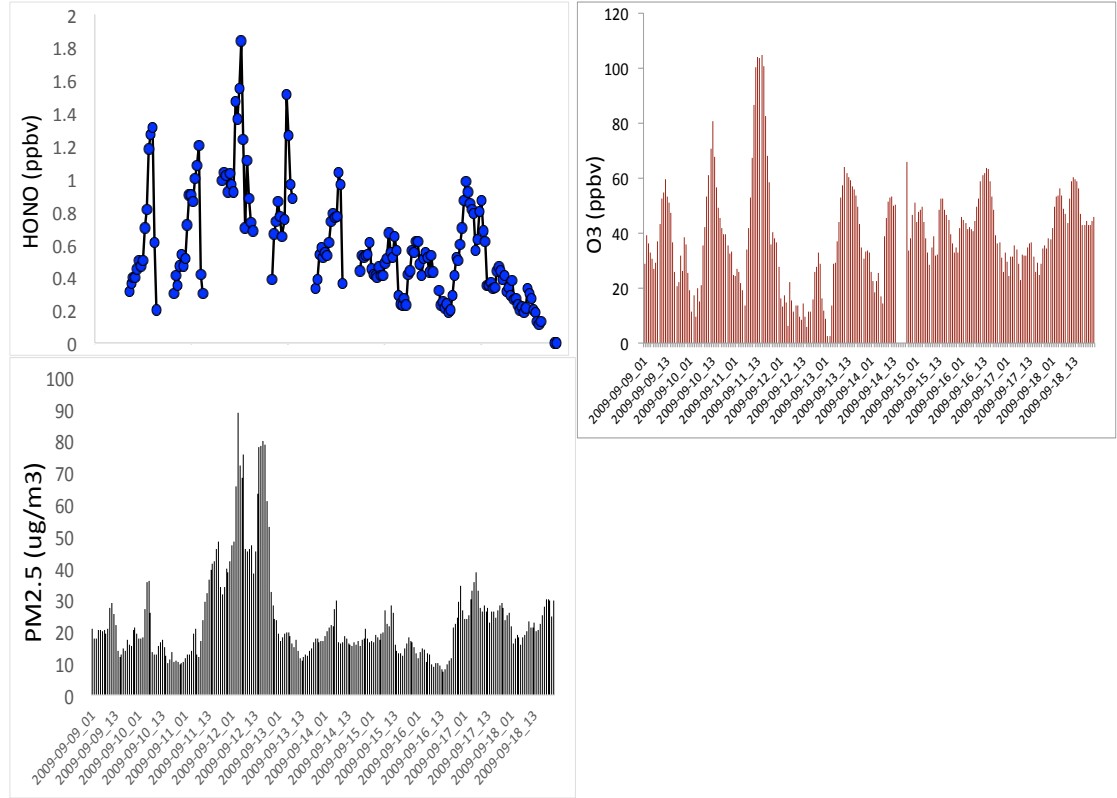

**Fig. 9.** The measured HONO (upper left panel), PM$_{2.5}$ concentrations (lower left
panel), and O$_3$ concentrations (upper right panel) in fall in Shanghai. It illustrates that
the high HONO concentrations were corresponded with high PM$_{2.5}$ concentrations.

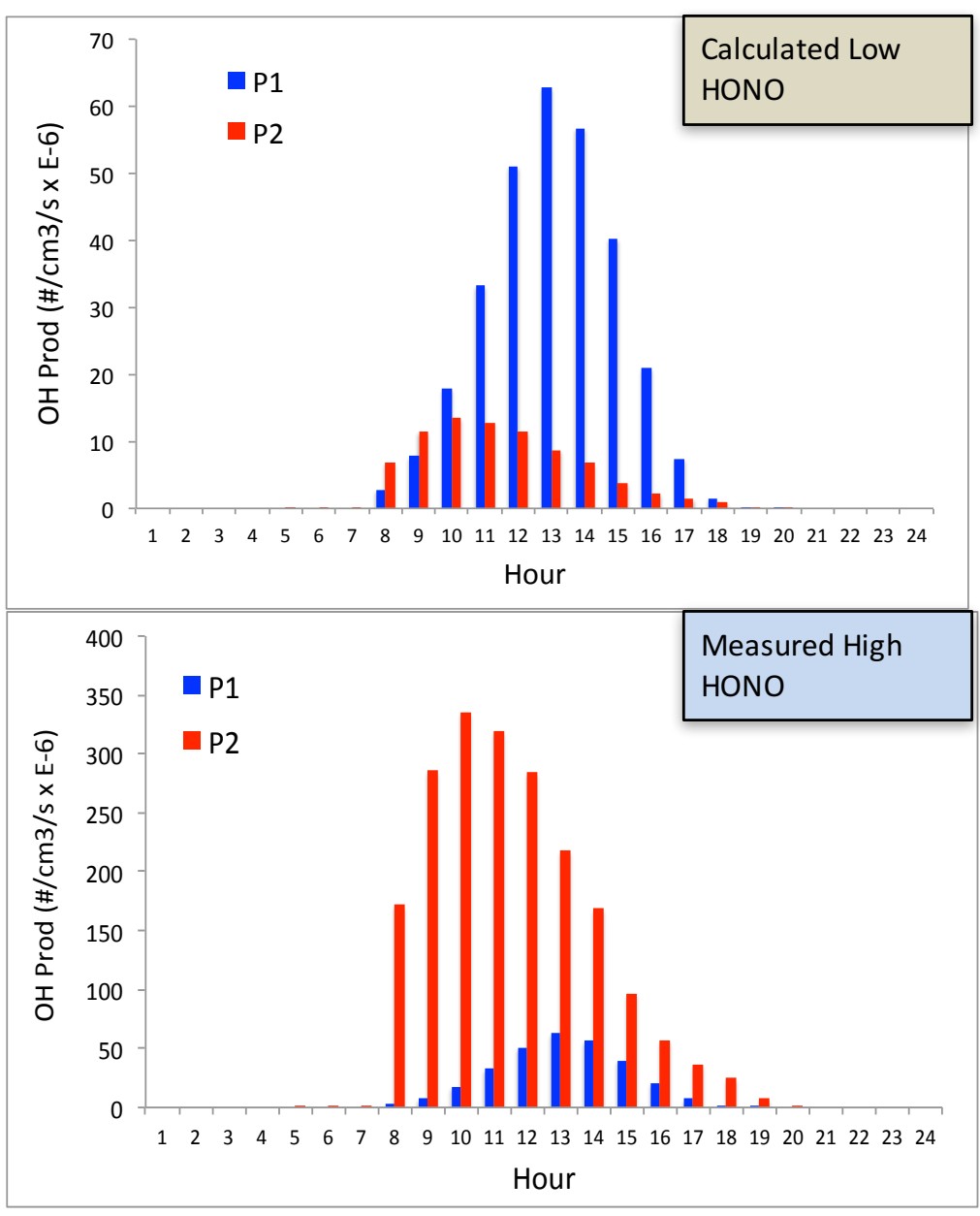

**Fig. 10.** The calculated OH production P(HOx) (#/cm³/s) by using the model calculated HONO (low concentrations) (in the upper panel) and by using the measured HONO (high concentrations) (in the lower panel). The red bars represent the calculation of the P1 term, and the red bars represent the calculation of the P2 term (OH production from HONO).

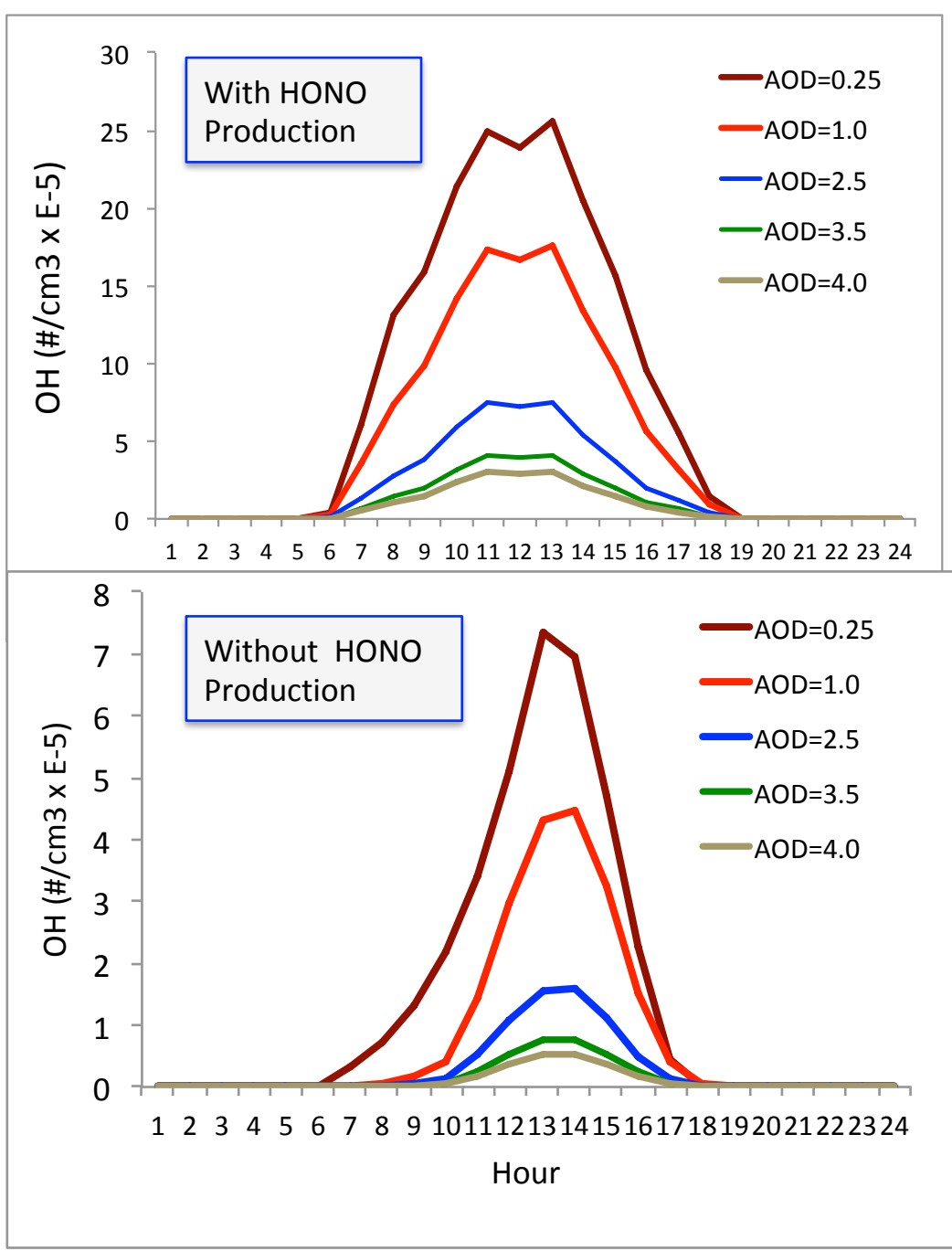

**Fig. 11.** The calculated OH concentrations (#/cm$^3$) with (upper panel) and
without (lower panel) HONO production of OH, under different aerosol
levels. Dark red (AOD=0.25), red (AOD=2.5) ), red (AOD=3.5) ), and red
(AOD=4.0).

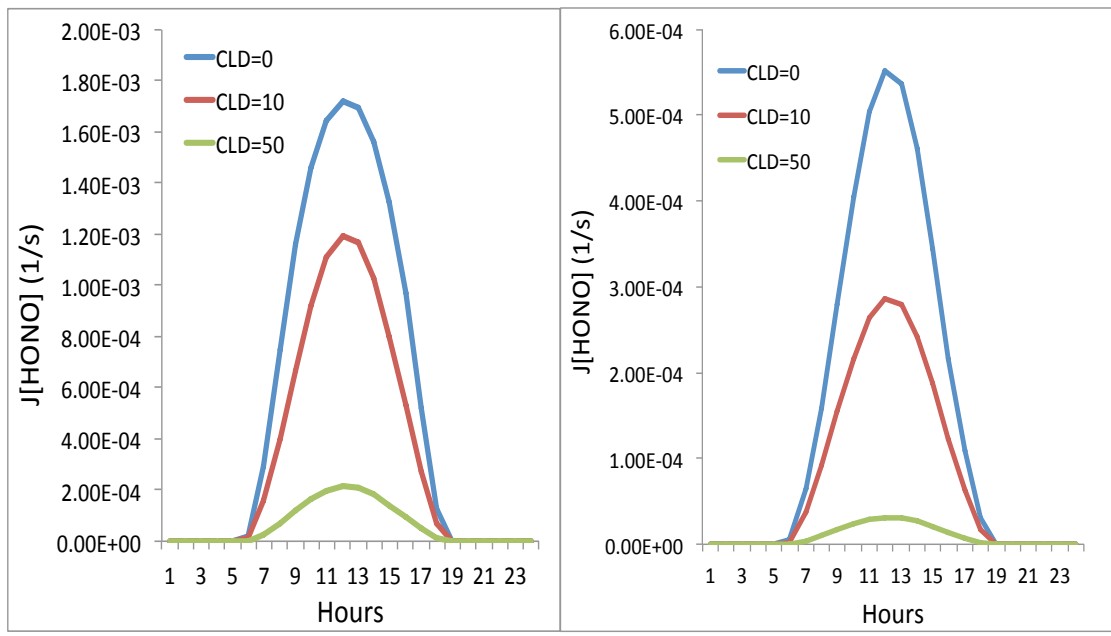

**Fig. 12.** The effect of cloud cover on the photolysis rate of HONO (J[HONO]). The
blue, red, and green lines represent the cloud water vapor of 0 (cloud-free), 10 (g/m$^3$ –
thin cloud), and 50 (g/m$^3$ – thick cloud), respectively. The left panel represents the
light aerosol condition, with AOD of 0.25, and the right panel represents the heavy
aerosol condition, with AOD of 2.5.

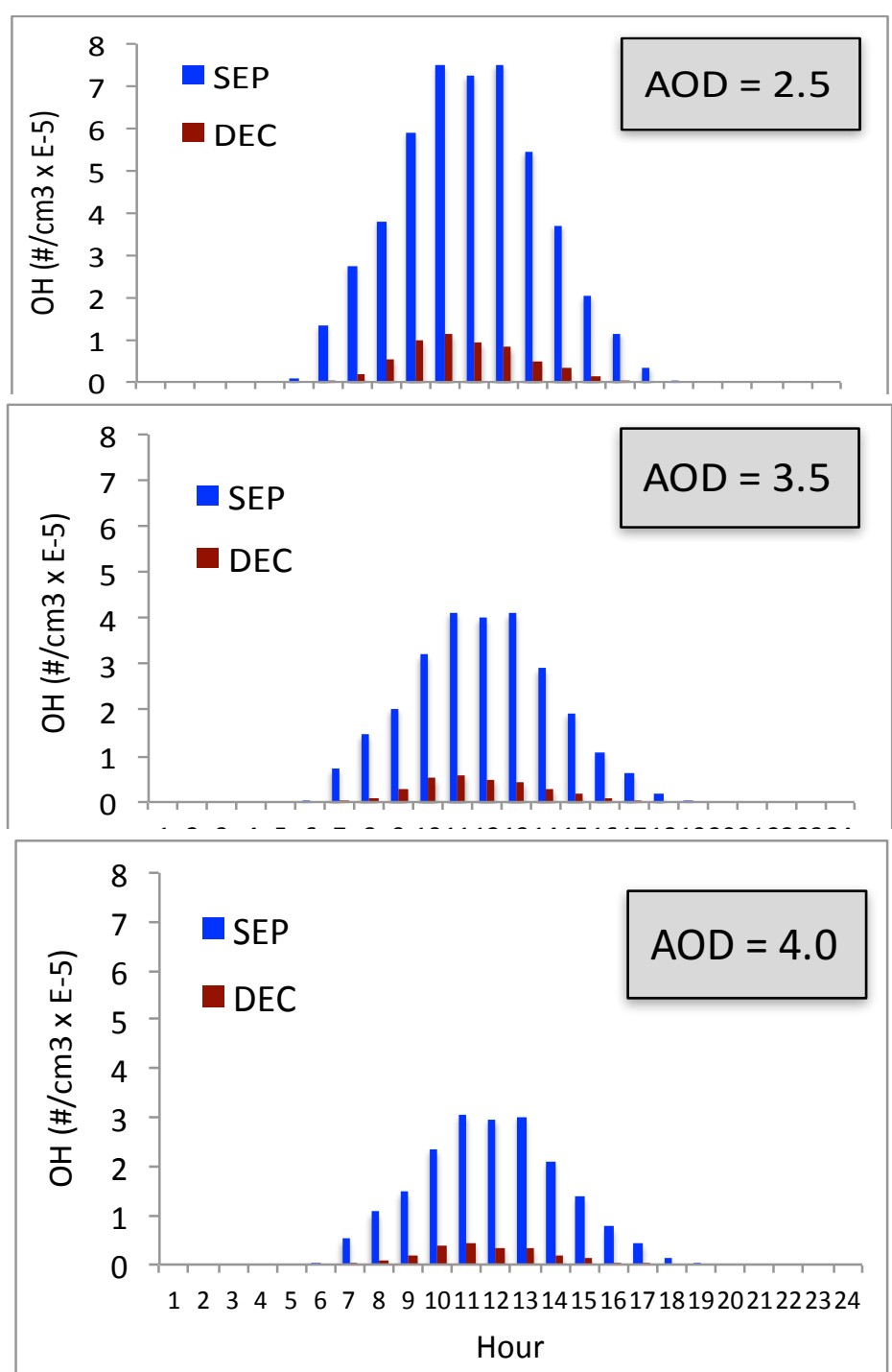

**Fig. 13.** The calculated OH concentrations in September (blue bars) and December (dark red bars), under different aerosol levels.