# Peer review of "Ozone enhancement due to photo-disassocation of nitrous acid in eastern China"

_Atmospheric Chemistry and Physics, 2019_

## Referee Comment (RC1) · Anonymous Referee #1 · 6 Jun 2019

This study could be a very meaningful work. The paper addressed the relevant scientific questions within the scope of ACP. This manuscript studied the possible reasons enhancing the ozone formation under high PM2.5 concentrations. It is not a very novel concept since some previous studies already reported the positive correlation between PM2.5 and ozone, and analyzed the underestimated HONO sources in China and other areas in the world. However, better understanding the mechanisms in different locations is scientifically significant in modeling studies. In addition, as the authors mentioned, the results bring important insights for control strategy of air pollution, because both PM2.5 and ozone are significant air pollutants in China.

There sever major concerns as follows:

(1) Both cloud and aerosol can affect the solar radiation. In order to separate these

two factors, especially for case studies, people usually will analyze the meteorological conditions during the measurement period, or only analyze the data under the cloud-free conditions. However, the authors of this manuscript never mentioned the cloud factor.

(2) Several important previous studies should be mentioned so that some conclusions from this manuscript can be more solid. For example (but not limited to), Zhang et al. (2016) already parameterized up-to-date HONO sources into WRF-Chem model such as the heterogeneous reactions on ground and aerosol surfaces, direct vehicle and vessel emissions, conversion of NO2 at the ocean surface, and emissions from soil bacteria. The modified WRF-Chem substantially reproduced the observed HONO levels, and greatly improved the ozone simulations. However, in this manuscript, the calculated HONO level was still very low in Fig. 6. More information about the WRF-Chem setup is needed. In addition, some other studies (e.g., Shi et al., 2015) already reported the positive correlation between aerosol and ozone. The ozone formation is also strongly dependent on the aerosol size and composition. The process might be a complex interaction between aerosols and photochemical reactions. For example, the scattering aerosol could considerably diffuse the solar radiation and enhance the flux density inside the boundary layer (He and Carmichael, 1999). Thus, the scattering aerosols may favor the ozone formation through increasing solar flux in the boundary layer (Shi et al., 2015). More discussions are needed in the manuscript.

Zhang, L., Wang, T., Zhang, Q., Zheng, J., Xu, Z., & Lv, M. (2016). Potential sources of nitrous acid (HONO) and their impacts on ozone: A WRF‐Chem study in a polluted subtropical region. Journal of Geophysical Research: Atmospheres, 121(7), 3645-3662.

Shi, C., Wang, S., Liu, R., Zhou, R., Li, D., Wang, W., ... & Zhou, B. (2015). A study of aerosol optical properties during ozone pollution episodes in 2013 over Shanghai, China. Atmospheric Research, 153, 235-249.

He, S., & Carmichael, G. R. (1999). Sensitivity of photolysis rates and ozone production in the troposphere to aerosol properties. Journal of Geophysical Research: Atmospheres, 104(D21), 26307-26324.

Generally, this manuscript presents a significant study; however, the analysis should be in more depth. The authors should give proper credit to related work, and clearly indicate this manuscript's original contribution. I would not recommend using a vague word (such as "low solar radiation") in the title.
* * *

---

## Referee Comment (RC2) · Anonymous Referee #2 · 10 Jun 2019

This work tried to explain the measured co-occurrence of high PM2.5 and O3 concentrations. The authors report that the high daytime HONO concentrations could be photo-dissociated to be OH radicals, which enhance the photochemical production of O3, although depressed solar radiation under heavy PM2.5 pollutions. It is an interesting scientific issues. However, the data and method in the manuscript do not support such a conclusion very well at this stage. My major concerns are listed as follows: (1) The authors mixed observations from Shanghai and Beijing to create an illusion. There are no observations to show high PM2.5-O3-HONO concentrations both at Shanghai and at Beijing. I just see high PM2.5-O3 during Oct.5-6, 2015 in Beijing and high PM2.5-HONO during September, 2009 in Shanghai. (2) Is the observed co-occurrence of high PM2.5 and O3 concentrations of statistical significance? Are the authors sure

it's (measurements during Oct.5-6) not a special case? (3) Could the authors make an effort to exclude the effects of precursor emissions (e.g., being sure that the VOCs/NOx ratios are not more beneficial for ozone production during Oct.5-6 than other days) and meteorological conditions (e.g., temperature and relative humidity; under low humidity, although the PM2.5 concentration is high, the solar radiation would not be depressed much) ? Moreover, there are no observations show the solar radiation are exactly depressed during Oct.5-6 in Beijing or September in Shanghai ? (4) If the authors insist the high PM2.5-O3-HONO mechanism, could this possible new mechanism be added to the WRF-Chem model for verification? (5) Discussion in sect.3.3: the conclusion (solar radiation in winter reaches a threshold level to prevent the OH chemical production, even by including the HONO production term) came too hastily without no direct evidence. Specific comments: (1) L167-169: there are no data to show the solar radiation are reduced (2) L185: same above (3) L188-190: same above (4) L199: "Chine" should be "China" (5) L201: removed "OH" (6) L218: what is "am" in O1D + am->O3P (7) L222: "Madronich and Flocke (1999)" should be "(Madronich and Flocke, 1999)" (8) L295-296: one of "P1" should be "P2"? (9) L298-299: one of "P1" should be "P2" ? (10) L241: What are possible sources of HONO?

---

## Author Comment (AC1) · 10 Jul 2019

Responses to Reviewers:

Reviewer 1:

We thank the reviewer for the careful reading of the manuscript and helpful comments. We have revised the manuscript following their suggestions as is described below.

This study could be a very meaningful work. The paper addressed the relevant scientific questions within the scope of ACP. This manuscript studied the possible reasons enhancing the ozone formation under high PM2.5 concentrations. It is not a very novel concept since some previous studies already reported the positive correlation between PM2.5 and ozone, and analyzed the underestimated HONO sources in China and

[Figure]

other areas in the world. However, better understanding the mechanisms in different locations is scientifically significant in modeling studies. In addition, as the authors mentioned, the results bring important insights for control strategy of air pollution, because both PM2.5 and ozone are significant air pollutants in China.

There sever major concerns as follows:

(1) Both cloud and aerosol can affect the solar radiation. In order to separate these two factors, especially for case studies, people usually will analyze the meteorological conditions during the measurement period, or only analyze the data under the cloud-free conditions. However, the authors of this manuscript never mentioned the cloud factor.

Thanks for the valuable comments of the reviewer. We have checked the meteorological condition (especially cloud condition) during the period of the case study (between Oct 5 and 6, 2015) in the Beijing region. It shows that there was close to the cloud free condition (see attached Fig-A1. Now it is Fig. 5 in the revised paper). In order to evaluate the effect of cloud, we made additional model runs (with thing and thick cloud conditions). The results show that clouds have important impact on the result of this study, and this study is more suitable for the cloud free conditions (see attached Fig-A2. Now it is Fig. 12 in the revised paper). The results show that the thin cloud (cloud cover in 2 km, with cloud water of 10 g/m3), could reduce the photolysis rate of HONO by about 40%, but the HONO could still remain important effects. However, with dense cloud condition (cloud covers at 2 and 3 km, with cloud water of 50 10 g/m3), the photolysis rate of HONO could reduce by 9-10 times by the cloud. In this case, adding photolysis rate of HONO cannot produce important effect on OH radicals and the production of O3. The above statements have been added in the revised manuscript.

(2) Several important previous studies should be mentioned so that some conclusions from this manuscript can be more solid. For example (but not limited to), Zhang et

al. (2016) already parameterized up-to-date HONO sources into WRF-Chem model such as the heterogeneous reactions on ground and aerosol surfaces, direct vehicle and vessel emissions, conversion of NO2 at the ocean surface, and emissions from soil bacteria. The modified WRF-Chem substantially reproduced the observed HONO levels, and greatly improved the ozone simulations. However, in this manuscript, the calculated HONO level was still very low in Fig. 6. More information about the WRF-Chem setup is needed. In addition, some other studies (e.g., Shi et al., 2015) already reported the positive correlation between aerosol and ozone. The ozone formation is also strongly dependent on the aerosol size and composition. The process might be a complex interaction between aerosols and photochemical reactions. For example, the scattering aerosol could considerably diffuse the solar radiation and enhance the flux density inside the boundary layer (He and Carmichael, 1999). Thus, the scattering aerosols may favor the ozone formation through increasing solar flux in the boundary layer (Shi et al., 2015). More discussions are needed in the manuscript.

Thanks for the valuable comments of the reviewer. We think that adding these previous studies will enhance the understanding of the highlights of our paper. The reviewer points that some recent versions of the WRF-Chem model add some missing HONO sources (surface emissions, conversion of NO2 at the ocean surface, etc.) can improve the HONO calculations (Shi et al., 2015). In our calculation, we only use the classical gas-phase chemistry to illustrate that the importance of these missing sources for the production of OH radicals. Adding these missing sources (there are not fully understand and remain a large uncertainty) could be a future work. In the revised paper, we add the above clarifications. We also add the reference of He and Carmichael (1999) to add their point that there maybe another factor that the ratio of the scattering and absorbing aerosols could be another factor to affect the relationship between aerosols and ozone. All the valuable references are included in the revised paper.

Zhang, L., Wang, T., Zhang, Q., Zheng, J., Xu, Z., & Lv, M. (2016). Potential sources of nitrous acid (HONO) and their impacts on ozone: A WRF/Chem study in a polluted

subtropical region. Journal of Geophysical Research: Atmospheres, 121(7), 3645-3662.

Shi, C., Wang, S., Liu, R., Zhou, R., Li, D., Wang, W., ... & Zhou, B. (2015). A study of aerosol optical properties during ozone pollution episodes in 2013 over Shanghai, China. Atmospheric Research, 153, 235-249. He, S., & Carmichael, G. R. (1999). Sensitivity of photolysis rates and ozone pro- duction in the troposphere to aerosol properties. Journal of Geophysical Research: Atmospheres, 104(D21), 26307-26324.

Generally, this manuscript presents a significant study; however, the analysis should be in more depth. The authors should give proper credit to related work, and clearly indicate this manuscript's original contribution. I would not recommend using a vague word (such as "low solar radiation") in the title.

Thanks. We change title from "Ozone formation under low solar radiation in eastern China" to "Ozone enhancement due to photo-disassociation of nitrous acid in eastern China"

[Figure]

Oct/05/15

Beijing Region

Oct/06/15

Beijing Region

Fig-A1. The cloud condition during the period of the case study (between Oct 5 and 6, 2015 in the Beijing region. The bright white color shows the cloud covers, and the grey white shows the haze covers. The Beijing region is under the heavy haze conditions during the period.

**Fig. 1.**

[Figure]

Fig-A2. The effect of cloud cover on the photolysis rate of HONO (J[HONO]). The blue, red, and green lines represent the cloud water vapor of 0 (cloud-free), 10 (g/m$^3$ – thin cloud), and 50 (g/m$^3$ – thick cloud), respectively. The left panel (A) represents the light aerosol condition, with AOD of 0.25, and the right panel (B) represents the heavy aerosol condition, with AOD of 2.5.

**Fig. 2.**

---

## Author Comment (AC2) · 10 Jul 2019

Responses to Reviewers:

Reviewer 2:

We thank the reviewer for the careful reading of the manuscript and helpful comments. We have revised the manuscript following their suggestions as is described below.

This work tried to explain the measured co-occurrence of high PM2.5 and O3 concentrations. The authors report that the high daytime HONO concentrations could be photo-dissociated to be OH radicals, which enhance the photochemical production of O3, although depressed solar radiation under heavy PM2.5 pollutions. It is an interesting scientific issues. However, the data and method in the manuscript do not support

such a conclusion very well at this stage.

My major concerns are listed as follows:

(1) The authors mixed observations from Shanghai and Beijing to create an illusion. There are no observations to show high PM2.5-O3-HONO concentrations both at Shanghai and at Beijing. I just see high PM2.5-O3 during Oct.5-6, 2015 in Beijing and high PM2.5-HONO during September, 2009 in Shanghai.

Thanks for pointing out this issue. The reason we chose the data by the following reasons. (1) Because the co-occurrence between O3 and PM2.5 are not always happened, it happens only in some episodes, especially in spring and fall. In winter, O3 and PM2.5 are actually anti-correlated due to low solar radiation (This also can see in Fig. 2 of the paper). It occurs under the following condition, (a) under cloud-free condition, (b) solar radiation is not too low, (c) during heavy aerosol pollutions in large cities in eastern China. Due to these limitations, it requires continuously measurements of O3 and PM2.5, and HONO concentrations. Recently, there are some continuously measurements of PM2.5, and O3 concentrations released by EPA of China. However, HONO measurements are not continuously measured, and we cannot find the HONO data with the period of co-occurrence between O3 and PM2.5. However, we do find some HONO measurements, which all shows that in all major Chinese cities in either fall or winter (Shanghai, Beijing, and Xian), the HONO concentrations were significant higher than other regions (see attached Fig-A1; Now in Fig. 8 of revised paper). For example, HONO concentrations reached highest in night, ranging from 1 to 2.5 ppbv in the morning at 6-9am. In daytime, the concentrations were lowest (ranging from 0.3 to 1.0 ppbv at12-18pm), but the concentrations were still significant higher than other regions, which could have significant effect on the production of OH radicals in daytime. As a result, we think that the high HONO is a common event in large cities in eastern China, especially in daytime. This high daytime high HONO is supported by the measurements in previous studies (Zhang et al. 2016; Huang et al. 2017). In this study, we make an assumption that the co-occurrence between O3 and PM2.5 occurred under

high HONO concentrations. From Fig.-A1, we also note that using this assumption may result in some uncertainties in estimating the effect of HONO on OH. For example, using the measured HONO in Xi'an and Beijing could produce 1-2 times higher OH production by photolysis of HONO than the result by using the measured data from Shanghai. In this case, we use the measured HONO from Shanghai to avoid the over estimate of the HONO effect, which can be considered as a low-limit estimation. The above statements are added in the revised paper.

(2) Is the observed co-occurrence of high PM2.5 and O3 concentrations of statistical significance? Are the authors sure it's (measurements during Oct.5-6) not a special case?

The co-occurrence of high PM2.5 and O3 concentrations was occurred in several cases in the past years. The attached Fig.-A2 shows some examples. Because it happened under some special conditions (see the reply in question 1), it most occurred in spring and fall seasons. (3) Could the authors make an effort to exclude the effects of precursor emissions (e.g., being sure that the VOCs/NOx ratios are not more beneficial for ozone production during Oct.5-6 than other days) and meteorological conditions (e.g., temperature and relative humidity; under low humidity, although the PM2.5 concentration is high, the solar radiation would not be depressed much)? Moreover, there are no observations show the solar radiation are exactly depressed during Oct.5-6 in Beijing or September in Shanghai?

Thanks for the valuable comments. We tried to find the available data, which is available during the period of Oct. 5 to 6, 2015. We do find some interesting data, which could answer the some comments of the reviewer. The additional data also helps to improve the quality of the paper. Fig.-A3 (now Fig. 5 in the revised paper) shows the cloud conditions in Beijing. During the period, there was close to the cloud free condiťionïijŇbut there was a very heavy aerosol layer. Fig.-A4 shows the relative humidity (RH) conditions. It shows that the RH (%) was generally higher than 60%, with a maximum of 95% during the period. As a result, the high aerosol concentrations companied

by high RH produced important effects on solar radiation. As shown in Fig.-A5 (now Fig. 6 in revised paper), the daytime averaged solar radiation was significantly reduced (about 40% reduction in Oct. 5-6 compared with the value of Oct. 8). We thanks the comments by the reviewer, these addition (figures and text) can significant enhance the quality of the paper.

(4) If the authors insist the high PM2.5-O3-HONO mechanism, could this possible new mechanism be added to the WRF-Chem model for verification?

Adding the high PM2.5-O3-HONO mechanism is a very challenge work, and could be another scientific work in the future. The major difficulty is that the causes (surface emissions or chemical transformations?) for the high HONO concentrations in large Chinese cities are not clearly understood. This could be a very interesting work in the future.

(5) Discussion in sect.3.3: the conclusion (solar radiation in winter reaches a threshold level to prevent the OH chemical production, even by including the HONO production term) came too hastily without no direct evidence.

Thanks for the comment. We agree with the reviewer that this conclusion is not very certain, and we re-write these sentences to soft the tone of this conclusion. In the revised paper, we change "When the solar radiation is in a very low level in winter, it reaches the threshold level to prevent the OH chemical production, even by including the HONO production of OH." to "Because the solar radiation is in a very low level in winter, adding the photolysis of HONO has smaller effect in winter than in fall, and OH remains low values by including the HONO production term."

Specific comments:

(1) L167-169: there are no data to show the solar radiation are reduced We add a new figure and text to show the solar reduction.

(2) L185: same above Answered in the above.

[Figure]

(3) L188-190: same above Answered in the above.

(4) L199: "Chine" should be "China" Corrected.

(5) L201: removed "OH" Corrected.

(6) L218: what is "am" in O1D + am->O3P am represents air mass in chemical reaction equations.

(7) L222: "Madronich and Flocke (1999)" should be "(Madronich and Flocke, 1999)" Corrected.

(8) L295-296: one of "P1" should be "P2"? Corrected.

(9) L298-299: one of "P1" should be "P2" ?  (10) L241:  What are possible sources of HONO? Corrected.  The possible sources of HONO could be surface sources or heterogeneous chemical reactions (but they are not fully understood at present).

Reference:

Zhang, L., Wang, T., Zhang, Q., Zheng, J., Xu, Z., & Lv, M. (2016). Potential sources of nitrous acid (HONO) and their impacts on ozone: A WRF/Chem study in a polluted subtropical region.  Journal of Geophysical Research: Atmospheres, 121(7), 3645-3662.

Huang, R. J., L. Yang, JJ Cao, QY Wang, X. Tie, et al., Concentration and sources of atmospheric nitrous acid (HONO) at an urban site in Western China. Sci. of Total Environ., 593-594, 165-172, doi.org/10.1016/j.scitotenv.2017.02.166, 2017.

[Figure]

[Figure]

**Fig. 6.** The measured HONO concentrations (ppbv) in three large cities in China. The red line was measured in Xi'An from 24 July to August 6, 2015. The blue line was measured in Shanghai from 9 to 18 September, 2009. The dark-red line was measured in Beijing from 1 to 27 January, 2014. The green line is calculated by the WRF-Chem model. The measurement in fall of Shanghai is applied to the calculation for the OH production of HONO.

**Fig. 1.**

Fig.-A2. The diurnal variations of $PM_{2.5}$ (blue line) and $O_3$ (red line), and $NO_2$ (green line) during the periods (from May 5 to May 7, 2013 (upper panel) and from Oct. 1 to Oct. 2, 2016 (lower panel)).

**Fig. 2.**

[Figure]

Fig-A3. The cloud condition during the period of the case study (between Oct 5 and 6, 2015 in the Beijing region. The bright white color shows the cloud covers, and the grey white shows the haze covers. The Beijing region is under the heavy haze conditions during the period.

**Fig. 3.**

Fig-A4. The measured relative humidity (RH) conditions between Oct. 5 and Oct.6, 2015.

**Fig. 4.**

[Figure]

Fig-A5. The measured solar radiation (W/m$^2$) from Oct. 3 to Oct. 9, 2015 in Beijing. The upper panel shows hourly values, and the lower panel shows the daytime averaged values.

**Fig. 5.**

---

## Author Response (AR2)

**Responses to Reviewers:**
**Reviewer 1:**
We thank the reviewer again for his/her careful reading of the manuscript and helpful comments.
We have revised the manuscript following the suggestions as is described below.
The authors made great efforts of revising the manuscript. However, the paper is still not
well written, and the conclusions were not convincingly supported by the data and method.
This is really an interesting scientific issue. I think there is still considerable more work
necessary to get the manuscript ready for publication at ACP. My major concerns are as
follows:
(1) The whole manuscript is based on the assumption that the co-occurrence of high ozone
and PM2.5 is under high HONO concentration. This assumption is highly possible to be true,
but it is lack of supportive measurement data. The authors have valuable HONO
measurements at three mega-cities including Beijing, Shanghai and Xi'An shown in Figure 8.
Since ozone and PM2.5 are routine measurement air pollutants, I would recommend
including them into the plot as well. Also, in Figure 8, since the measurement time is
different, I do not think they are comparable. I recommend separating Figure 8 into three
subplots by including ozone and PM2.5, and each subplot is for each city. So that the
assumption should be more solid.
Thanks for the constructive suggestion. We have separated Fig. 8 to 3 subplots. Fig. 8a shows
the measured PM2.5 and O3, along with the measured HONO in Beijing. Fig. 8b shows the
measured PM2.5 and O3, along with the measured and calculated HONO in Shanghai.     Fig.
8c shows the measured PM2.5 and O3, along with the measured HONO in Xi'an.    All figures
show that there were co-occurrences of high O3 and PM2.5, from late spring to early fall,
along with high HONO concentrations. These figures make the assumption to be more solid.
We have added the corresponding text in the revised version.
(2) The authors still did not state the set up of the WRF-Chem simulation, e.g. the gas-phase
mechanism used in the model? The authors need to at least briefly explain why the HONO
calculated by WRF-Chem is much lower than the observation. I think the model only
consider the HONO source with NO+OH only right? Also, how could the authors compare one
WRF-Chem modeling result to observations at three different cities during three
measurement time periods? All of those statement and comparison are not rigorous. Please
revise.
To address the comments of the reviewer, we add more details regarding the chemical
scheme of the WRF-Chem (the version which we used). We adding that "**The version of the**
**WRF-Chem model is based on the version developed by Grell et al. (2015), and is improved**
**mainly by Tie et al. (2007) and Li et al. (2011). The chemical mechanism chosen in this**
**version of WRF-Chem is the RADM2 (Regional Acid Deposition Model, version 2) gas-phase**
**chemical mechanism. For the calculation of HONO, only the gas-phase chemistry of**
**OH+NO is included to calculate HONO concentrations. As shown in Fig. 8, the calculated**
**HONO concentrations are significantly smaller than the measured HONO values in eastern**
**China, suggesting that in addition to the gas-reaction, there are missing HONO sources**
**(surface sources or others). Because these missing sources are not fully understood and large**
**uncertainty is remained, in the following calculation, we compare the OH concentrations**
**due to both calculated HONO (without the missing sources) and the measured HONO**
**concentrations to illustrate the importance of these missing sources for the production of OH**
**radicals and to suggest that further study to better understand the missing sources is an**
**urgent scientific issue".**
(3) Some conclusions and rationales are not rigorous. For example:

Line 278-279: Unless the authors show the error bars, this conclusion is not solid.
We revise this statement
Line 281-287: see my major concern (1).
According to the reviewer's suggestion, we make 3 subplots (see answer 1)
Line 289-295: If it is possible, it would be very helpful to include ozone measurement into
Figure 9 as well.
Following the reviewer's comment, we add O3 measurement in Fig. 9.
(4) The literature is not cited properly:
Line 100-102: the mixed regime for ozone formation is missed in the statement.
Added.
Line 130: Shi et al. (2015) never talked about "several potential HONO sources, including
surface emissions, conversion of NO 2 at the ocean surface, etc., and adding these sources
can improve the calculated HONO concentrations." These conclusions are from Zhang et al.
(2016).
Corrected.
Line 266: see my comments above, wrong citation.
Corrected.
(5) The paper is not very well written and organized. There are numerous typos and
grammar errors. Please carefully review the whole manuscript and revise them accordingly.
I listed some as follows, but not limited to:
Line 35: only "fall"? It seems the authors mentioned both "late spring and fall" in the
manuscript?
Corrected. Changed to "from late spring to early fall" in all manuscript.
Line 56: here is "spring and fall"? Please be consistent through the whole manuscript.
Corrected.
Line 99: grammar error - "… are becomes …" Please revise.
Corrected.
Line 121: is it just "fall" or "late spring and fall"? Please be consistent through the whole.
Corrected.
Line 145 and 219: two section 2? Please revise.
Corrected. Also for the following numbers of sections.
Line 174-176: the sentence is redundant. Consider the following:
"The heavy aerosol concentrations play important roles to reduce solar radiation, causing
the reduction of O3 formation."
Thanks. The sentence is changed according to the suggestion of the reviewer.
Line 176: there is no Fig. 3a. Please indicate the upper panel as (a) in the plot or in the figure
capital.
Corrected.
Line 187: now the seasons include "late spring, summer, and early fall" instead of "late
spring and fall". I am very confused. Please be consistent about the seasons through the
whole manuscript.

Thanks for point out this typo. We checked all text, and changed to a consistent word "from
late spring to early fall".
Line 204-205: the sentence is redundant. Consider the following:
"both PM2.5 and O3 are severe air pollutants in eastern China."
Thanks. The sentence is changed according to the suggestion of the reviewer.
Line 207-217: Good!
Line 219 and Line 145: two section 2? Please revise.
Corrected.
Line 225: you mean "the surface solar radiation", not "the surface of solar radiation" right?
Corrected.
Line 236-237: "It can be expressed as"
Corrected.
Line 297-298: the sentence is redundant. Consider the following:
"the high HONO concentrations in daytime become a significant source of OH radicals."
Thanks. The sentence is changed according to the suggestion of the reviewer.
Line 339: it is "P2" not "P1" right?
Corrected.
Line 363 and Line 380: two section 3.3.
Corrected.
Line 384: "Figure 10 shows the OH concentrations in September and December"? What does
this mean? I thought Figure 10 shows a sensitivity study of OH production P using measured
and modeled HONO. Do I understand this correctly? Please revise.
Sorry. It should be Fig. 13 not Fig. 10. Corrected.
Line 412-413: "a double peak of PM2.5 and O3"? It sounds like for each pollutant, there is a
double peak. You mean "a co-occurrence of high PM2.5 and O3 concentrations"?
Thanks. We change this sentence to "a co-occurrence of high PM2.5 and O3 in some cases"
Line 413 and 432: only "fall" season?
Corrected.
Line 440: Delete "Because"
Corrected.

[revised manuscript text omitted]

Xuexi Tie 8/8/2019 9:14 AM

Xuexi Tie 8/8/2019 9:14 AM

Xuexi Tie 8/8/2019 9:14 AM

Xuexi Tie 8/8/2019 9:45 AM

Xuexi Tie 8/8/2019 9:45 AM

Xuexi Tie 8/8/2019 9:15 AM
**Moved (insertion) [1]**

[Figure]

Unknown

**Fig. 8b.** The measured HONO concentrations (ppbv) and the $PM_{2.5}$ and $O_3$ daily
concentrations in Shanghai. The upper panel shows the measured daily concentrations of
$PM_{2.5}$ and $O_3$ in 2015. The dark-red line was measured in Shanghai from 9 to 18 September,
2009. The green line was calculated by the WRF-Chem model.

[Figure]

[Figure]

**Fig. 8c.** The measured HONO concentrations (ppbv) and the $PM_{2.5}$ and $O_3$ daily
concentrations in Xi'an. The upper panel shows the measured daily concentrations of $PM_{2.5}$
and $O_3$ in 2015. The red line was measured HONO in Xi'An from 24 July to August 6, 2015.

**Unknown**

**Xuexi Tie 8/8/2019 9:56 AM**
Deleted: three large cities in China. The red line was measured in Xi'An from 24 July to August 6, 2015. The blue line was measured in Shanghai from 9 to 18 September, 2009. The dark-red line was measured in Beijing from 1 to 27 January, 2014. The green line is calculated by the WRF-Chem model. The measurement in fall of Shanghai is applied to the calculation for the OH production of HONO. ... [1]

**Xuexi Tie 8/8/2019 9:15 AM**
Moved up [1]: The dark-red line was measured in Beijing from 1 to 27 January, 2014.

**Xuexi Tie 8/8/2019 10:17 AM**

[Figure]

**Unknown**

[Figure]

**Fig. 9.** The measured HONO (upper left panel), PM$_{2.5}$ concentrations (lower left panel), and O$_3$
concentrations (upper right panel) in fall in Shanghai. It illustrates that the high HONO
concentrations were corresponded with high PM$_{2.5}$ concentrations.

Unknown

Xuexi Tie 8/8/2019 10:17 AM
Xuexi Tie 8/8/2019 10:18 AM

[Figure]

**Fig. 10.** The calculated OH production P(HOx) (#/cm³/s) by using the model calculated HONO (low concentrations) (in the upper panel) and by using the measured HONO (high concentrations) (in the lower panel). The red bars represent the calculation of the P1 term, and the red bars represent the calculation of the P2 term (OH production from HONO).

[Figure]

**Fig. 11.** The calculated OH concentrations ($\#/cm^3$) with (upper panel) and without (lower panel) HONO production of OH, under different aerosol levels. Dark red (AOD=0.25), red (AOD=2.5) ), red (AOD=3.5) ), and red (AOD=4.0).

[Figure]

**Fig. 12.** The effect of cloud cover on the photolysis rate of HONO (J[HONO]). The blue, red, and
green lines represent the cloud water vapor of 0 (cloud-free), 10 (g/m$^3$ – thin cloud), and 50 (g/m$^3$
– thick cloud), respectively. The left panel (A) represents the light aerosol condition, with AOD of
0.25, and the right panel (B) represents the heavy aerosol condition, with AOD of 2.5.

[Figure]

**Fig. 13.** The calculated OH concentrations in September (blue bars) and December (dark red bars),
under different aerosol levels.

---

## Author Response (AR3)

**Responses to the Editor:**

We thank the Editor again for his careful reading of the manuscript and helpful comments. We have revised the manuscript following the suggestions as is described below.

1. Please enhance the quality of the figures by increasing both the resolution and the sizes of the figure legends.

According to the Editor's suggestion, we re-plot the figures with better quality and large sizes of the figure legends.

2. The following studies are concerned with the impact of carbonaceous aerosol on air pollution that may be referred to in the introduction:
Jia R., M. Luo, Y. Liu, Q. Z. Zhu, S. Hua, C. Q. Wu and T. B. Shao, 2019: Anthropogenic Aerosol Pollution over the Eastern Slope of the Tibetan Plateau. Advances in Atmospheric Sciences, 2019, 36(8): 847-862.

Zhu Q., Y. Liu, R. Jia, S. Hua, T. Shao, B. Wang, 2018: A numerical simulation study on the impact of smoke aerosols from Russian forest fires on the air pollution over Asia. Atmospheric Environment, 182,263-274.

According to the Editor's suggestion, we have added these 2 references in the introduction and references.

Xuexi Tie 8/13/2019 12:06 PM

[revised manuscript text omitted]

Xuexi Tie 8/13/2019 12:01 PM

Unknown

Xuexi Tie 8/13/2019 12:03 PM

Unknown

Xuexi Tie 8/13/2019 2:36 PM

Xuexi Tie 8/13/2019 12:04 PM

Xuexi Tie 8/13/2019 12:05 PM

[Figure]

**Fig. 4.** The diurnal variations of PM$_{2.5}$ (blue line) and O$_3$ (red line), and NO$_2$ (green line) during a fall period (from Oct. 5 to Oct. 6, 2015). It shows that with high PM$_{2.5}$ condition, there was a strong O$_3$ diurnal variation.

Unknown

Xuexi Tie 8/13/2019 12:05 PM

[Figure]

Xuexi Tie 8/13/2019 2:36 PM

[Figure]

[Figure]

**Fig. 5.** The cloud condition during the period of the case study (between Oct 5 and 6,
2015 in the Beijing region. The bright white color shows the cloud covers, and the
grey white shows the haze covers. The Beijing region is under the heavy haze
conditions during the period.

[Figure]

**Fig. 6.** The measured solar radiation (W/m$^2$) from Oct. 3 to Oct. 9, 2015 in Beijing. The upper panel shows hourly values, and the lower panel shows the daytime averaged values.

[Figure]

**Fig. 7.** The effect of aerosol levels with AOD = 0.25 (black line), AOD = 2.5 (red
line), AOD = 3.5 (blue line), and AOD = 4.0 (green line) on the $O_3$ photolysis
calculated by the TUV model in October at middle-latitude.

[Figure]

Xuexi Tie 8/13/2019 2:13 PM

Unknown

Xuexi Tie 8/13/2019 2:38 PM

[Figure]

[Figure]

**Fig. 8a.** The measured HONO concentrations (ppbv) and the $PM_{2.5}$ and $O_3$ daily
concentrations in Beijing. The upper panel shows the measured daily concentrations
of $PM_{2.5}$ and $O_3$ as shown in Fig.2. The dark-red line was measured HONO in Beijing
from 1 to 27 January, 2014.

Xuexi Tie 8/13/2019 2:14 PM

Unknown
Roman, Font color: Text 1
Xuexi Tie 8/13/2019 2:38 PM

[Figure]

[Figure]

**Fig. 8b.** The measured HONO concentrations (ppbv) and the $PM_{2.5}$ and $O_3$ daily
concentrations in Shanghai. The upper panel shows the measured daily concentrations
of $PM_{2.5}$ and $O_3$ in 2015. The dark-red line was measured in Shanghai from 9 to 18
September, 2009. The green line was calculated by the WRF-Chem model.

Unknown

Xuexi Tie 8/13/2019 2:38 PM

[Figure]

[Figure]

**Fig. 8c.** The measured HONO concentrations (ppbv) and the PM$_{2.5}$ and O$_3$ daily
concentrations in Xi'an. The upper panel shows the measured daily concentrations of
PM$_{2.5}$ and O$_3$ in 2015. The red line was measured HONO in Xi'An from 24 July to
August 6, 2015.

Unknown

Xuexi Tie 8/13/2019 2:38 PM

[Figure]

**Fig. 9.** The measured HONO (upper left panel), PM$_{2.5}$ concentrations (lower left
panel), and O$_3$ concentrations (upper right panel) in fall in Shanghai. It illustrates that
the high HONO concentrations were corresponded with high PM$_{2.5}$ concentrations.

[Figure]

Xuexi Tie 8/13/2019 2:16 PM

Unknown

Xuexi Tie 8/13/2019 2:38 PM

[Figure]

**Fig. 10.** The calculated OH production P(HOx) (#/cm³/s) by using the model
calculated HONO (low concentrations) (in the upper panel) and by using the
measured HONO (high concentrations) (in the lower panel). The red bars
represent the calculation of the P1 term, and the red bars represent the calculation
of the P2 term   (OH production from HONO).

[Figure]

**Fig. 11.** The calculated OH concentrations (#/cm$^3$) with (upper panel) and without (lower panel) HONO production of OH, under different aerosol levels. Dark red (AOD=0.25), red (AOD=2.5) ), red (AOD=3.5) ), and red (AOD=4.0).

Xuexi Tie 8/13/2019 2:39 PM

[Figure]

**Fig. 12.** The effect of cloud cover on the photolysis rate of HONO (J[HONO]). The
blue, red, and green lines represent the cloud water vapor of 0 (cloud-free), 10 (g/m³ –
thin cloud), and 50 (g/m³ – thick cloud), respectively. The left panel represents the
light aerosol condition, with AOD of 0.25, and the right panel represents the heavy
aerosol condition, with AOD of 2.5.

Xuexi Tie 8/13/2019 2:17 PM

[Figure]

Unknown

Xuexi Tie 8/13/2019 2:39 PM

Xuexi Tie 8/13/2019 2:18 PM

Xuexi Tie 8/13/2019 2:18 PM

[Figure]

[Figure]

Xuexi Tie 8/13/2019 2:18 PM

**Unknown**

**Fig. 13.** The calculated OH concentrations in September (blue bars) and
December (dark red bars), under different aerosol levels.

Xuexi Tie 8/13/2019 2:39 PM